# Telomere-to-telomere *Citrullus* super-pangenome provides direction for watermelon breeding

Yilin Zhang [1,2,5], Mingxia Zhao [1,5], Jingsheng Tan [1,5], Minghan Huang[2,5], Xiao Chu[1], Yan Li[1], Xue Han[1], Taohong Fang[1], Yao Tian[1], Robert Jarret [3], Dongdong Lu [1], Yijun Chen [1], Lifang Xue[1], Xiaoni Li[1], Guochen Qin [1], Bosheng Li[1], Yudong Sun[4], Xing Wang Deng [1,2], Yun Deng [1] ✉, Xingping Zhang [1] ✉ & Hang He [1,2] ✉

To decipher the genetic diversity within the cucurbit genus *Citrullus*, we generated telomere-to-telomere (T2T) assemblies of 27 distinct genotypes, encompassing all seven *Citrullus* species. This T2T super-pangenome has expanded the previously published reference genome, T2T-G42, by adding 399.2 Mb and 11,225 genes. Comparative analysis has unveiled gene variants and structural variations (SVs), shedding light on watermelon evolution and domestication processes that enhanced attributes such as bitterness and sugar content while compromising disease resistance. Multidisease-resistant loci from *Citrullus amarus* and *Citrullus mucosospermus* were successfully introduced into cultivated *Citrullus lanatus*. The SVs identified in *C. lanatus* have not only been inherited from *cordophanus* but also from *C. mucosospermus*, suggesting additional ancestors beyond *cordophanus* in the lineage of cultivated watermelon. Our investigation substantially improves the comprehension of watermelon genome diversity, furnishing comprehensive reference genomes for all *Citrullus* species. This advancement aids in the exploration and genetic enhancement of watermelon using its wild relatives.

Watermelon (*Citrullus lanatus* (Thunb.) Matsum. & Nakai) is an economically important crop grown throughout the world. Cultivated watermelons, even when collected from different geographical regions, typically exhibit low genetic diversity[1]. This lack of genetic diversity for important traits within the *C. lanatus* gene pool has resulted in a bottleneck that may be impeding watermelon improvement and led to a shifted focus toward characterizing and using the genetic variations within watermelon's crop wild relatives (CWRs). The genus

*Citrullus* comprises six additional species alongside *C. lanatus*, *Citrullus amarus* and *Citrullus mucosospermus*, which are semi-wild and locally harvested for their edible flesh or seeds, while the remaining *Citrullus* species exhibit unique adaptive traits crucial for genetic enhancement and understanding watermelon evolution. Among the six watermelon CWRs, *C. amarus*, *Citrullus ecirrhosus*, *Citrullus naudinianus* and *Citrullus rehmii* are indigenous to the Namib-Kalahari region, while *C. mucosospermus* hails from West Africa and *Citrullus*

[1]National Key Laboratory of Wheat Improvement, Peking University Institute of Advanced Agricultural Sciences, Shandong Laboratory of Advanced Agricultural Sciences at Weifang, Weifang, China. [2]State Key Laboratory of Protein and Plant Gene Research, School of Advanced Agricultural Sciences and School of Life Sciences, Peking University, Beijing, China. [3]Plant Genetic Resource Unit, Griffin, GA, USA. [4]Vegetable Research and Development Center, Huaiyin Institute of Agricultural Sciences of Xuhai Region in Jiangsu, Huai'an, China. [5]These authors contributed equally: Yilin Zhang, Mingxia Zhao, Jingsheng Tan, Minghan Huang. ✉e-mail: yun.deng@pku-iaas.edu.cn; xingping.zhang@pku-iaas.edu.cn; hang.he@pku-iaas.edu.cn

*colocynthis* from northern Africa and beyond. *C. colocynthis*, *C. amarus* and *C. mucosospermus* have been used in breeding programs to broaden the genetic base of *C. lanatus*[2]. However, the plant genetic resources of *C. rehmii*, *C. ecirrhosus*, *C. naudinianus* and *C. lanatus* subsp. *cordophanus* that are available for research are quite limited. Increased availability of plant materials of these wild species will likely lead to the discovery of new alleles associated with disease resistance, abiotic stress tolerance and variation for the presence or enhancement of human health-promoting compounds such as cucurbitacin[3,4] and citrulline[4,5].

Watermelon genomic studies have so far been largely limited to cultivated accessions of *C. lanatus*[6,7], *C. colocynthis*[2] and *C. lanatus* subsp. *cordophanus*[1], and recently the telomere-to-telomere (T2T) genome G42 raised the single watermelon genome assembly to a new level[7]. However, pangenome analyses offer opportunities to identify genetic diversity in a wider gene pool that includes representative examples of multiple genotypes. Pangenomes in various crop genera have demonstrated the power of identifying genetic variants, discovering and identifying functional genes and empowering the genetic improvement of crops[8–11]. To examine and exploit the diversity present within the genomes of CWRs, a genus-level strategy based on the construction of a 'super-pangenome' was suggested[12], as demonstrated in refs. 13,10. The genus-level super-pangenome provides access to previously untapped genetic variation present within the CWRs and facilitates exploration of the dispensable portion of the genome[14].

In this study, we construct a genus-wide super-pangenome by de novo assembling 27 T2T gap-free genomes that include all seven watermelon species and *C. lanatus* subsp. *cordophanus*. Comparative analyses revealed the landscape of the genus' genomic content, domestication history and the distribution of structural variations (SVs). These data serve to empower the discovery of genes within the genomes of wild *Citrullus* species that might be used to increase biotic and abiotic resistance and resilience in the cultivated watermelon crop.

## Results

### T2T assemblies of 27 diverse accessions from seven species

To create a super-pangenome representing the genetic diversity and genome structure of the entire *Citrullus* genus, we strategically selected 27 representative accessions (Fig. 1a–c), including one with an exiting T2T gap-free genome, based on phylogenetic relationships and the geographical distributions of 429 accessions (Fig. 1d). This group contains all seven *Citrullus* species (Supplementary Table 1). The phenotypes of the 27 accessions are highly diverse (Supplementary Fig. 1a–c). In addition to the representative accessions of each species, we also included some disease-resistance accessions (Supplementary Note 1).

To survey the *Citrullus* genomes, we first generated Illumina sequencing data to an average depth of 62× for the 27 accessions (Supplementary Table 2). The results revealed that these materials were diploid, with low heterozygosity and a repeat rate of around 55% (Supplementary Table 1). Subsequently, we generated 799 Gb of high-fidelity (HiFi) sequencing data for the 27 accessions. This amounted to approximately an 81× depth relative to the estimated 380 Mb size of the watermelon genome (Supplementary Table 2). Each

genome was then de novo assembled. The N50 length of the contigs of the 27 whole-genome assemblies ranged from 30.1 to 39.8 Mb, averaging 33.3 Mb (Table 1). By leveraging Oxford Nanopore Technologies (ONT) ultra-long reads (average depth of 85×) and high-throughput chromosome conformation capture (Hi-C) technology (average depth of 699×), we assembled 27 representative T2T genomes (Supplementary Table 2). Contigs of seven accessions were corrected, ordered and oriented using Hi-C sequencing data, while those of the remaining accessions were oriented using the respective species reference genome (Supplementary Fig. 1d–i). The newly assembled genomes exhibited good collinearity with the previously released G42 genome[7], as shown in the Circos (Supplementary Fig. 2) and dot-plot diagrams (Supplementary Fig. 3). The assembled genomes contained an average of two gaps, indicative of the high level of accuracy and continuity of the assemblies. Gaps were filled using the HiFi and ONT ultra-long read sequencing data as described for the G42 genome assembly[7], and the quality of each gap-filled region was assessed (Supplementary Table 3). The telomeres of PI 595203 (*C. mucosospermus*), PI 537300 (*C. colocynthis*), PI 482276 (*C. amarus*), PI 632755 (*C. colocynthis*), PI 652554 (*C. colocynthis*), PI 673135 (*C. ecirrhosus*) and PI 670011 (*C. rehmii*) were filled using seven-base telomere repeats (CCCTAAA) as a sequence query (Supplementary Fig. 2). The enhancement of telomere assembly was substantially facilitated by the use of HiFi and ONT sequencing data (Supplementary Table 4). Our integrated assembly strategy generated *Citrullus* reference genomes of high precision and T2T integrity. The final assembled genome sizes ranged between 361.3 Mb and 413.6 Mb, averaging 375.2 Mb (Table 1). Additionally, we assembled the chloroplast and mitochondrial genomes for each of the 27 watermelon accessions. These averaged 156.9 kb and 622.2 kb in length, respectively (Supplementary Table 5).

Assembly quality was evaluated from several perspectives (Supplementary Note 1). Repetitive DNA comprised about 56.2% of each genome, with Gypsy elements being the most common. For gene annotation, RNA sequencing (RNA-seq) of multiple tissues at different developmental stages was performed, predicting an average of 24,698 protein-coding genes per genome (Supplementary Note 1). In conclusion, these comprehensive pangenome assemblies and accompanying gene resources provide a solid foundation for the further exploration and use of the whole-genome gene repository in the context of watermelon biology and breeding endeavors.

### Detection of the centromere locations at chromosomes

The candidate centromere regions of all chromosomes in 27 watermelon accessions were identified (Supplementary Table 11) and confirmed by the Hi-C heatmap (Supplementary Fig. 5a). By comparing the centromere monomer-based phylogeny with the whole-genome gene-based phylogeny, it can be observed that although *C. naudinianus* is distantly related to *C. lanatus* at the whole-genome level, its centromere sequences (except for chromosome 5) are closer to *C. lanatus*. Additionally, at the whole-genome level, *C. amarus* is more closely related to *C. lanatus* than to *C. colocynthis*. However, in the evolutionary relationships of centromere sequences, *C. colocynthis* is closer to

**Fig. 1 | Genetic diversity and phenotypes of watermelon accessions for T2T or gap-free assembly. a–c**, The highly diverse fruit/seed phenotypes of seven watermelon species and interspecific hybrids−*C. naudinianus*: 1 PI 596694; *C. colocynthis*: 2–5 PI 525081, PI 632755, PI 652554 and PI 537300; *C. rehmii*: 6 PI 670011; *C. ecirrhosus*: 7 PI 673135; *C. amarus*: 8–12 PI 482276, PI 296341-FR, PI 271769, PI 189225 and RCAT 055816; *C. mucosospermus*: 13–14 PI 532732 and PI 595203; *C. lanatus* landrace: 15–20 PI 254622, HeiShanRen, PI 381740, DaBanHongZiGua, PI 288522 and SanBaiGua; *C. lanatus* cultivar: 21–27 ShiHong No. 2, Sugarlee, Charleston Gray, Calhoun Gray, Allsugar, G42 and PKR6. The 28–31 interspecific hybrids of *C. lanatus* cultivar and *C. colocynthis*; 32–34 interspecific hybrids of *C. lanatus* cultivar, *C. ecirrhosus* and *C. mucosospermus*. Scale bar corresponds to 1 cm. **a**, Whole fruit; **b**, longitudinal section of fruit;

**c**, seeds. **d**, Neighbor-joining phylogenetic tree of 429 accessions inferred from genome-wide SNPs. The colors of branches in the tree indicate different species−*C. lanatus* (teal), *C. mucosospermus* (purple), *C. amarus* (light blue), *C. rehmii* (red), *C. ecirrhosus* (green), *C. colocynthis* (dark blue) and *C. naudinianus* (orange). In Figures 2c,d, 3a,b and 4b and Supplementary Figures 1a–c, 2, 3, 6a, 8a, 10, 11, 13, 14 and 15, consistent color coding has been adopted to represent the species of the accessions. The 27 accessions used for de novo assembly are indicated with stars in the phylogenetic tree. **e**, Types and percentages of different TE families detected in the seven groups genome. **f**, A neighbor-joining phylogenetic tree was constructed using the top eight monomers from each of the 26 watermelon accessions. Colors of branches in the tree indicate different groups. Clusters 1–4 represent similar monomers.

*C. lanatus*. This suggests that in *C. amarus* and *C. colocynthis* species, the centromere regions of some chromosomes have evolved independently (Supplementary Fig. 5b). We observed a different transposable element (TE) composition between the centromere region of chr02, chr03, chr04 and chr06 and that of other chromosomes (Supplementary Table 12), which may have led to their diverse evolutionary

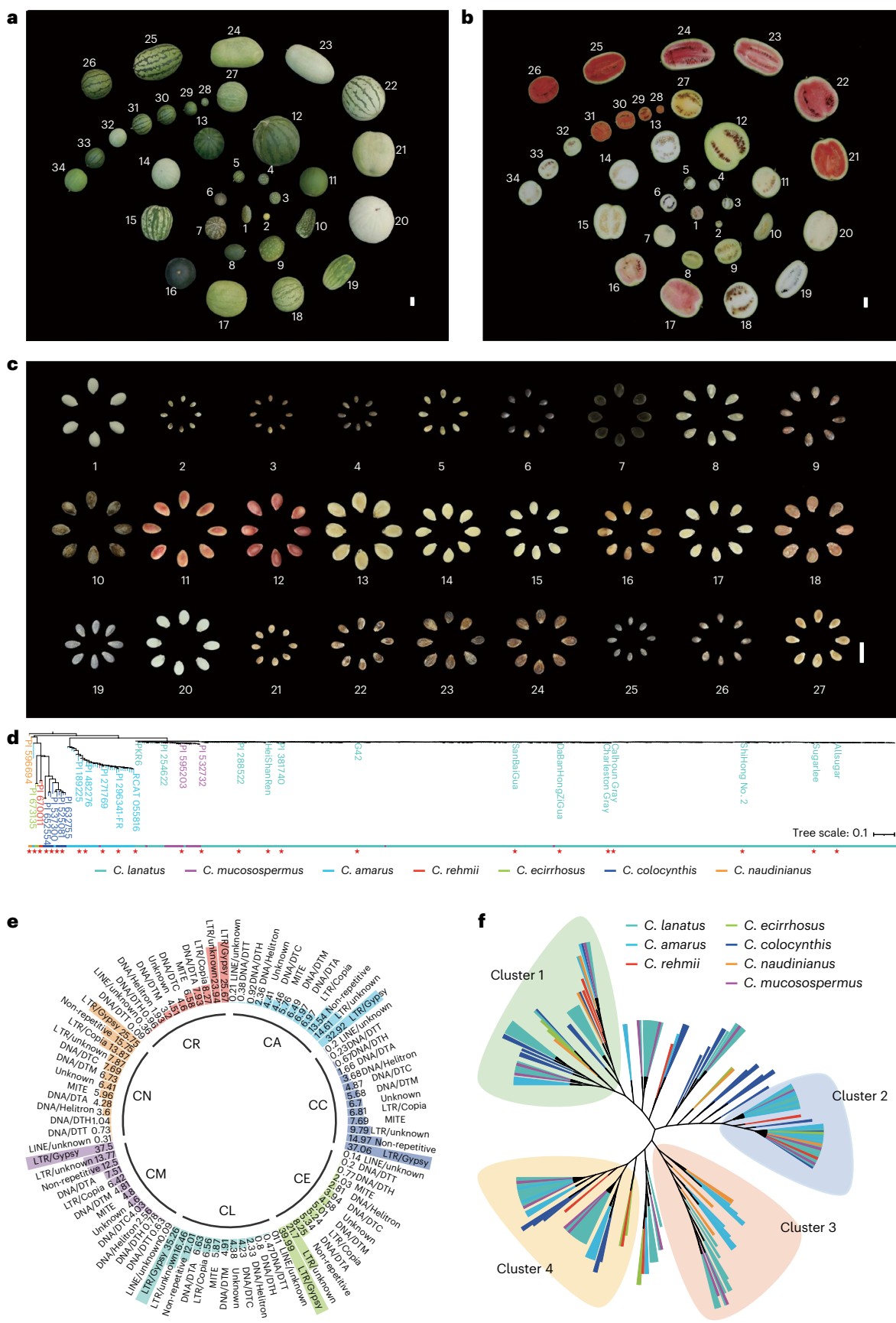

**Table 1 | Summary of the assembly and annotation of 27 watermelon genomes**

| Sample | Contig N50 | Gap number | Telomere | BUSCO | QV | LAI | Assembly size (Mb) | TE content | Gene number | Subpopulation |
|---|---|---|---|---|---|---|---|---|---|---|
| G42 | 32.5M | 0 | 22 | 99.10% | 76.9 | 10.88 | 369.3M | 56.74% | 24,205 | *C. lanatus* |
| SanBaiGua | 33.0M | 0 | 22 | 99.10% | 50.9 | 9.84 | 370.4M | 55.75% | 25,493 | *C. lanatus* |
| Calhoun Gray | 32.2M | 0 | 22 | 99.10% | 72.8 | 9.46 | 369.6M | 55.47% | 24,642 | *C. lanatus* |
| Charleston Gray | 32.4M | 0 | 22 | 99.10% | 63.3 | 9.46 | 369.0M | 55.67% | 23,825 | *C. lanatus* |
| HeiShanRen | 33.1M | 0 | 22 | 99.10% | 70.8 | 10.04 | 371.2M | 56.03% | 25,562 | *C. lanatus* |
| PI 254622 | 30.1M | 0 | 22 | 99.10% | 65.4 | 10.04 | 373.7M | 55.74% | 24,976 | *C. lanatus* |
| AllSugar | 31.5M | 0 | 22 | 99.10% | 75.4 | 8.95 | 369.4M | 55.88% | 25,498 | *C. lanatus* |
| ShiHong No. 2 | 32.7M | 0 | 22 | 99.10% | 68.4 | 9.61 | 368.9M | 55.74% | 24,666 | *C. lanatus* |
| Sugarlee | 35.6M | 0 | 22 | 99.10% | 72.9 | 9.66 | 369.3M | 55.58% | 24,935 | *C. lanatus* |
| PI 381740 | 30.1M | 0 | 22 | 99.10% | 68.9 | 9.71 | 369.6M | 55.60% | 24,228 | *C. lanatus* |
| DBHZGua | 32.4M | 0 | 22 | 99.20% | 67.9 | 10.08 | 369.9M | 55.64% | 23,973 | *C. lanatus* |
| PKR6 | 32.9M | 0 | 22 | 99.10% | 66.8 | 9.72 | 371.4M | 55.45% | 23,758 | *C. lanatus* |
| PI 288522 | 32.5M | 0 | 22 | 99.20% | 70.0 | 9.9 | 368.4M | 55.85% | 24,092 | *C. lanatus* |
| PI 595203 | 31.9M | 0 | 22 | 99.00% | 72.9 | 9.89 | 371.0M | 55.68% | 25,232 | *C. mucosospermus* |
| PI 532732 | 33.5M | 0 | 22 | 99.10% | 67.2 | 9.45 | 370.7M | 55.98% | 25,530 | *C. mucosospermus* |
| PI 482276 | 38.8M | 0 | 22 | 99.00% | 75.3 | 4.65 | 381.5M | 57.45% | 24,011 | *C. amarus* |
| PI 296341-FR | 38.4M | 0 | 22 | 99.00% | 69.0 | 11.34 | 381.0M | 57.50% | 24,897 | *C. amarus* |
| RCAT 055816 | 35.0M | 0 | 22 | 99.00% | 67.5 | 4.86 | 380.2M | 57.42% | 23,369 | *C. amarus* |
| PI 189225 | 31.5M | 0 | 22 | 99.10% | 67.56 | 4.89 | 377.8M | 57.11% | 23,551 | *C. amarus* |
| PI 271769 | 34.6M | 0 | 22 | 99.10% | 73.9 | 10.93 | 378.0M | 56.21% | 25,887 | *C. amarus* |
| PI 537300 | 33.1M | 0 | 22 | 98.50% | 73.0 | 12.02 | 380.0M | 56.90% | 24,081 | *C. colocynthis* |
| PI 652554 | 30.6M | 0 | 22 | 99.00% | 69.2 | 8.86 | 361.3M | 54.20% | 24,208 | *C. colocynthis* |
| PI 525081 | 32.4M | 0 | 22 | 99.00% | 72.2 | 10.84 | 378.8M | 56.74% | 24,247 | *C. colocynthis* |
| PI 632755 | 33.5M | 0 | 22 | 99.10% | 73.2 | 10.55 | 378.6M | 56.30% | 24,418 | *C. colocynthis* |
| PI 670011 | 39.8M | 0 | 22 | 99.00% | 72.3 | 11.04 | 413.6M | 60.96% | 26,969 | *C. rehmii* |
| PI 673135 | 34.3M | 0 | 22 | 98.70% | 74.3 | 9.01 | 402.4M | 59.59% | 24,101 | *C. ecirrhosus* |
| PI 596694 | 32.0M | 0 | 22 | 99.10% | 65.3 | 10.16 | 364.5M | 54.78% | 25,277 | *C. naudinianus* |

relationships. A cluster with four candidate centromeric tandem repeats, which occurred in the majority of the 27 watermelon accessions genomic sequences, was identified (Supplementary Table 13).

One of the major candidate centromere tandem repeats (cluster 1) is identical to Cr2 in G42 (ref. 7), and this repeat is the most abundant in all accessions except for PKR6 (*C. lanatus*), PI 652554 (*C. colocynthis*) and PI 537300 (*C. colocynthis*). The frequency of genes and TE repeats, such as long terminal repeat (LTR) (Gypsy and Copia) and a versatile transposon hAT family (DNA/DTA), is relatively high in the centromere region. Among the repeats predicted in the 27 watermelon accessions, the largest portion comprises LTR (average 56.6%), predominantly Gypsy (average 33.45%) and Copia (average 7.73%) elements (Fig. 1e). The phylogenetic analysis of representative monomers from the 27 watermelon accessions revealed four main clusters. Of note, the *C. colocynthis*, *C. rehmii* and *C. naudinianus* monomers showed distinct patterns of clustering, indicating differences in their centromeric tandem repeats when compared to other species (Fig. 1f).

**Watermelon super-pangenome construction and analysis**
To explore the genomic landscape of the gene family, the phylogenetic tree built on the presence–absence variation (PAV) of gene families from the 28 accessions (Supplementary Note 2) is close to previous classifications of the genus *Citrullus* using single-nucleotide polymorphisms (SNPs; Supplementary Fig. 6a). The representativeness of the accessions was estimated by observing the variation in the number

of the pangenome, core genome and dispensable genome genes each time a new genome was added (Fig. 2a). A simulation analysis randomizing the order of watermelon accessions suggests that the pangenome constructed in this study is closed (Fig. 2a). The size of the pangenome of the 28 accessions is approximately 1.5 times that of each individual genome, adding 11,225 gene families relative to the cultivated watermelon. Compared to the study discussed in ref. 14, our pangenome elucidated an additional 8,736 new gene families, with 4,913 gene families being contributed by the *C. rehmii*, *C. ecirrhosus* and *C. naudinianus* species.

Overall, an average of 42.78%, 8.80%, 45.87% and 2.55% of genes in each genome were classified as core genes, softcore genes, dispensable genes and private genes, respectively (Fig. 2b). In each species, an average of 52.72%, 43.81% and 3.47% of genes were identified as core genes, dispensable genes and private genes, respectively (Supplementary Fig. 6b). Dispensable and private genes account for most of the phenotypic diversification between species. The presence–absence distribution of core genes and dispensable genes among genomes (Fig. 2c,d) and species (Supplementary Fig. 6c,d) accounts for differences in gene PAV of up to about 40%, suggesting a high level of plasticity in the watermelon genome.

A total of 96.3% of core genes and 88.0% of softcore genes contain annotated InterPro domains, substantially higher than that of dispensable and private genes (55.6% and 32.4%, respectively; Supplementary Fig. 7a). The average expression level of core genes is substantially higher

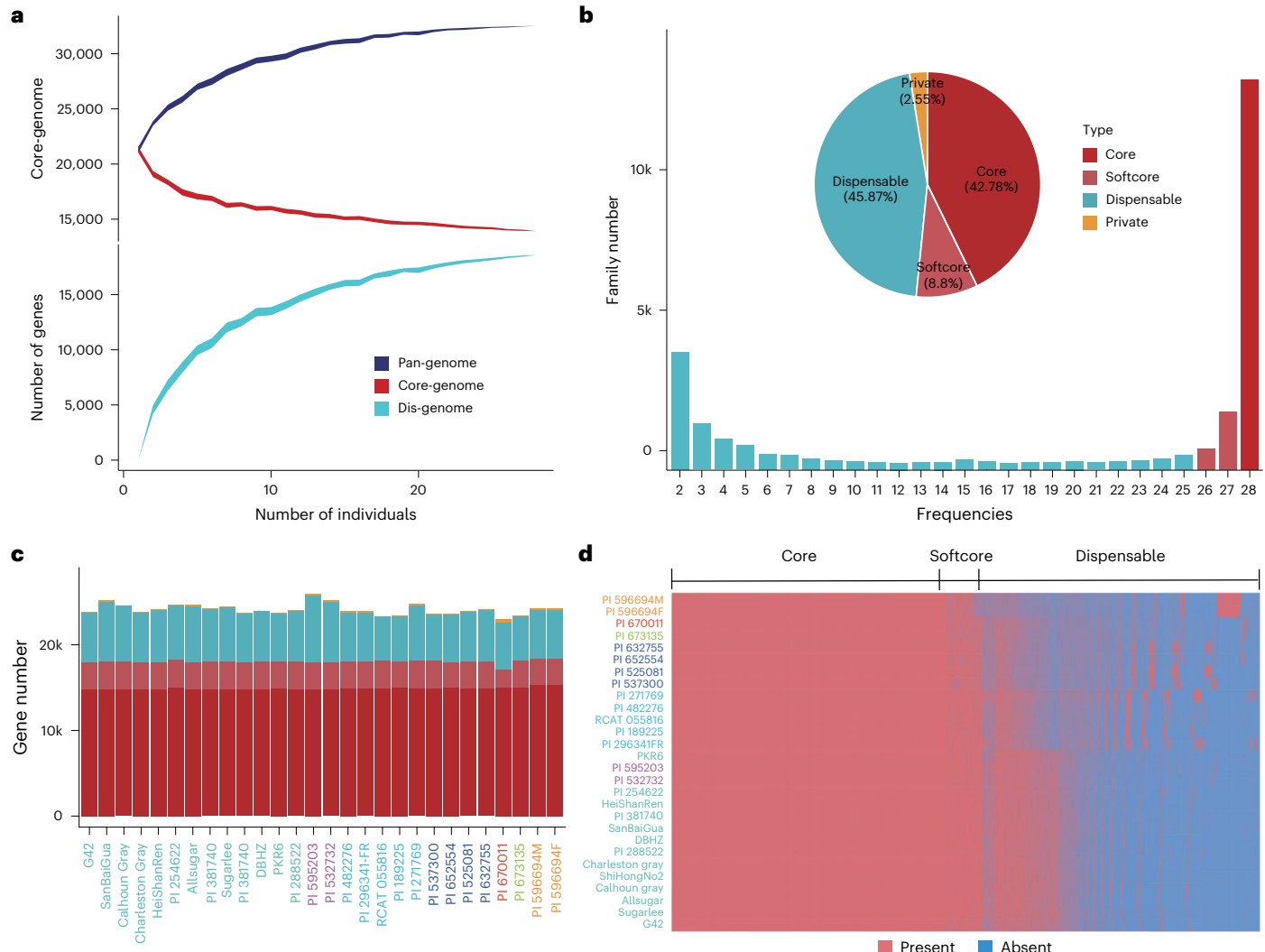

**Fig. 2 | Pangenome and core-genome analyses of 28 watermelon accessions.**
**a**, Analyses of the pangenome and core-genome models' sizes reveal a relationship with pangenome composition, signifying a closed pangenome. The breadth of the curves represents the 100× of replication. **b**, Compositions of core, softcore, dispensable, private genes of the accession-level pangenomes. **c**, Gene numbers of each gene type in 28 watermelon genomes. **d**, Mapping the landscape of presence–absence across nonredundant gene families for 28 watermelon accessions.

than that of dispensable and private genes (Supplementary Fig. 7b). Nucleotide diversity (π) in core genes was lower than in dispensable genes (Supplementary Fig. 7c). These results collectively support the premise that core genes are involved in fundamental biological processes, dispensable genes exhibit greater genetic variation and may have a role in environmental adaptation, whereas private genes may have important roles in determining individual species morphological characteristics.

**SV characterization and graph-based genome**

SV can have a greater impact on genomic polymorphism and functional gene variation than SNPs[15]. By aligning the 27 new genomes to the G42 reference genome, SVs (>20 bp) were classified into the following five categories: deletions, insertions, duplications, inversions and translocations. All the 461,987 nonredundant SVs comprise 217,076 (46.99%) deletions, 239,698 (51.88%) insertions, 3,122 (0.67%) duplications, 1,278 (0.28%) inversions and 813 (0.18%) translocations (Fig. 3a). When these data were compared to G42, an average of 81,132 SVs (ranging from 6,390 to 217,199) per accession were identified. These SVs affected an average of 150 Mb of genomic sequence per accession (ranging from 9.93 Mb to 364.5 Mb; Supplementary Fig. 8a). To check the accuracy of SV identification, 57 large SVs with an average length of 26,418 bp, including 31 deletions and 26 insertions, were validated

by PCR amplification (Supplementary Fig. 8b and Supplementary Table 14). By mapping all assembly variations and their collinearity with the reference genome, we established a comprehensive SV landscape. This landscape revealed a narrow genetic diversity within cultivated watermelons, but notably greater genetic variation in the wild watermelons (Fig. 3b). Inversion 'hotspots' were identified (Supplementary Fig. 8c). It has been reported that large inversions can suppress recombination by reducing crossing-over[9]. On Chr11 of the PKR6 (*C. lanatus*) accession, two large inversions (4.8 Mb and 2.5 Mb) inherited from *C. amarus* are retained (Supplementary Table 15). Large inversions have been reported in the constructed genetic maps[16] and have been shown to substantially reduce recombination frequency[16,17]. Introgression genes located close to these regions from *C. amarus* through backcross breeding may lead to severe linkage drag of unexpected phenotypes. We identified and annotated genes both within and outside these large (>100 kb) inversions/translocations (Supplementary Table 16), revealing that genes predominantly pertain to cellulose synthesis, as demonstrated by substantial enrichment in pathways such as GO:0016759 and GO:0030244 (Supplementary Table 17). Furthermore, five key genes, including *ClG42_04g0123000* and *ClG42_03g0079100*, have been reported to relate to traits like seed size and fruit sweetness[18,19], providing valuable insights into understanding the functions of these SVs.

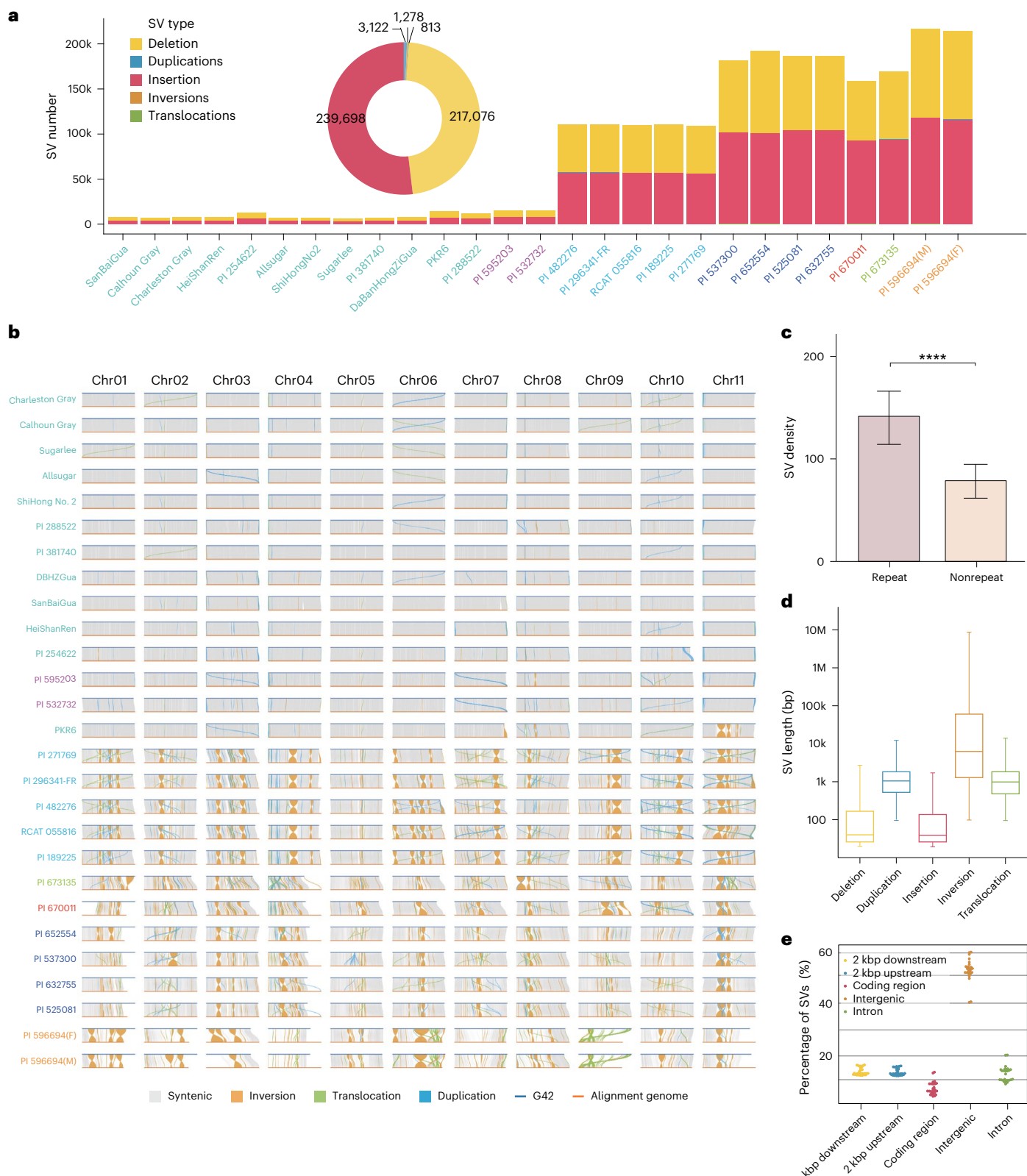

**Fig. 3 | The landscape of the genetic SVs among the 28 watermelon genomes. a**, Comparison of various SV types in each genome relative to the G42 reference genome. **b**, SyRI-derived comparative genomic visualization map illustrating the synteny and rearrangements among 27 gap-free genomes in relation to the G42 reference accession. **c**, Density of SVs within repetitive and nonrepetitive genomic regions at 500-kb intervals (*n* = 27). Data are presented as mean values ± s.e.m. Significance was ascertained using two-sided Fisher's exact test, with ****$P$ < 0.0001. **d**, Size distributions for different types of SVs (deletion = 217,076, duplication = 3,122, insertion = 239,698, inversion = 1,278 and translocation = 813) between G42 and other watermelon genomes. The edges and the centerlines of the boxes represent the IQR and medians, with the whiskers extending to the most extreme points (1.5× IQR). **e**, Percentage of SVs sharing overlap with different genomic regions between G42 and other watermelon genomes. IQR, interquartile range.

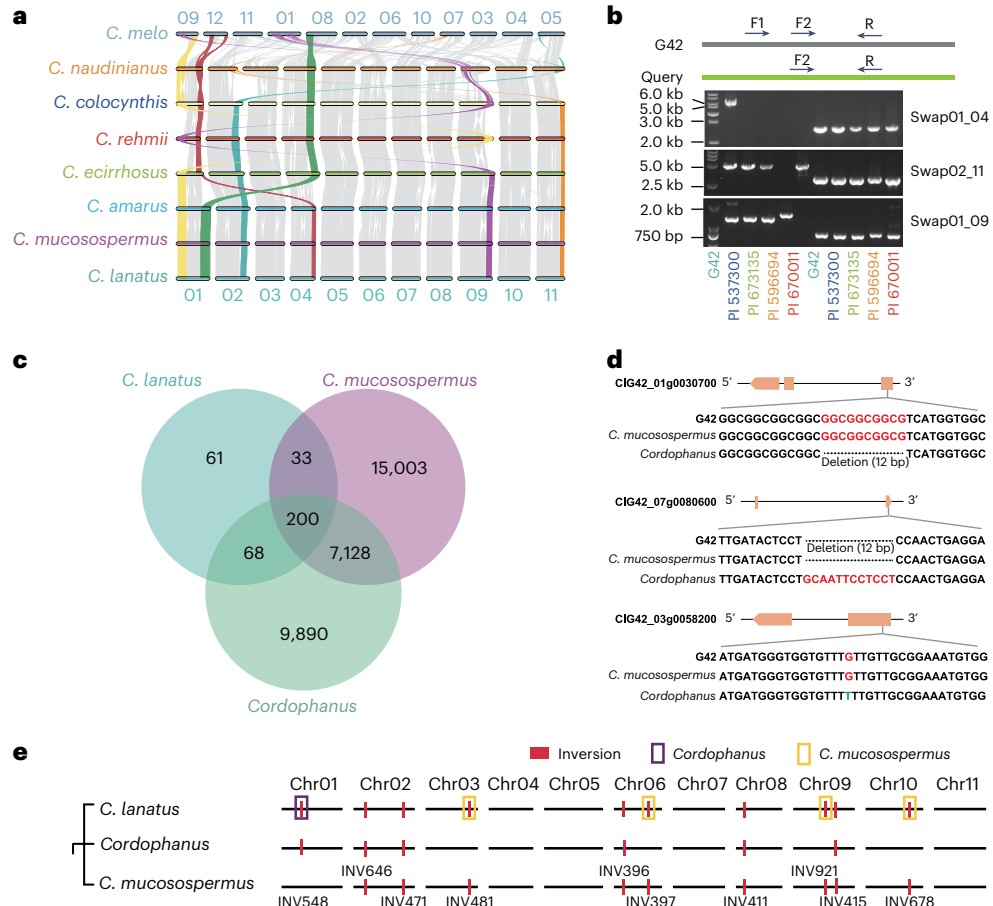

**Fig. 4 | Divergence among watermelon species and the origin of domesticated watermelons. a**, Syntenic gene blocks among the seven watermelon species and *Cucumis melo*. **b**, PCR validation of chromosomal rearrangement. The first five bands were amplified using primer pairs F1 and R, while the last five bands were amplified using primer pairs F2 and R. **c**, The Venn diagram illustrates the identification and intersections of core SVs, within *C. lanatus* and two potential ancestors of cultivated watermelon. **d**, Variations in three genes related to agronomic traits. **e**, Genome-based map of large inversions within cultivated watermelon and its two probable ancestors. Purple boxes denote inversions within *C. lanatus* that accord with those in *C. lanatus* subsp. *cordophanus*, while yellow boxes highlight inversions consistent with those in *C. mucosospermus*. The PCR validation of chromosomal rearrangement was repeated three times.

The SVs in watermelon tend to be enriched in regions of repetitive DNA and types of deletions and insertions (Fig. 3c,d), a pattern consistent with previous studies on soybean[13]. We observed that an average of 27.5% of SVs overlapped with the 2 kbp regions upstream or downstream of genes. An average of 7.5% of SVs caused changes in amino acid coding, potentially contributing to gene functional diversity (Fig. 3e). We expanded SVs to the population scale and developed a web-based database for the graph-based pangenome (www.watermelondb.cn; Supplementary Note 3). These findings indicate that the SVs in the super-pangenome reflect large structural changes during the evolution of the cultivated watermelon and its related species, which deepens our understanding of the genomic and phenotypic changes of the *Citrullus* evolution.

## Divergence among species and the origin of watermelon

Chromosomal rearrangements are of considerable interest in understanding adaptation and speciation because they form barriers to gene flow between related species[20]. Three major chromosomal rearrangements among species in *Citrullus* were identified and validated (Fig. 4a,b and Supplementary Table 18). As noted in ref. 14, we identified a substantial interchromosomal rearrangement involving chr01 and chr04 in *C. colocynthis* compared to three other *Citrullus* species (*C. lanatus, C. mucosospermus* and *C. amarus*). These alterations in the chromosome structure might contribute to reproductive isolation, impacting hybrid fertility and reducing interspecific recombination[16,17,21–25], ultimately

leading to the differentiation of *Citrullus* species. To investigate the conservation and variation of the three-dimensional genome among different watermelon species, we identified A and B compartments using a 50 kb resolution matrix. The results showed that A and B compartments are relatively conserved among watermelon species (Supplementary Fig. 9a). It has been reported that variations in A and B compartments are closely related to genomic SVs[26]. Among different types of SVs, we observed variation for a 4.5 Mb inversion that resulted in a change in A and B compartments (Supplementary Fig. 9b).

Previous reports suggested that the likely ancestor of watermelon is *C. lanatus* subsp. *cordophanus*, a wild watermelon type from Sudan in Northeast Africa[1]. We discovered an explosive increase in SVs between the wild watermelon (*C. amarus*) and the cultivated watermelon (*C. lanatus*). In contrast, there was no significant difference in SVs between *C. mucosospermus* and the cultivated watermelon (Fig. 3a). We identified 362 SVs that occurred in *C. lanatus*, 33 of which were inherited from *C. mucosospermus*, 68 from *C. lanatus*. subsp. *cordophanus* and 200 that were common to both *C. mucosospermus* and *C. lanatus* subsp. *cordophanus* (Fig. 4c). SVs in three genes among G42, *C. mucosospermus* and *C. lanatus* subsp. *cordophanus* are displayed in Fig. 4d. Among them, *ClG42_07g0080600* was identified by selective sweeps, while *ClG42_01g0030700* and *ClG42_03g0058200* were reported to be associated with fruit size and seed coat color quantitative trait locus (QTLs), respectively[27,28]. *C. lanatus* and *C. mucosospermus* carry the same haplotype, while *C. lanatus* subsp. *cordophanus* carries an alternative one.

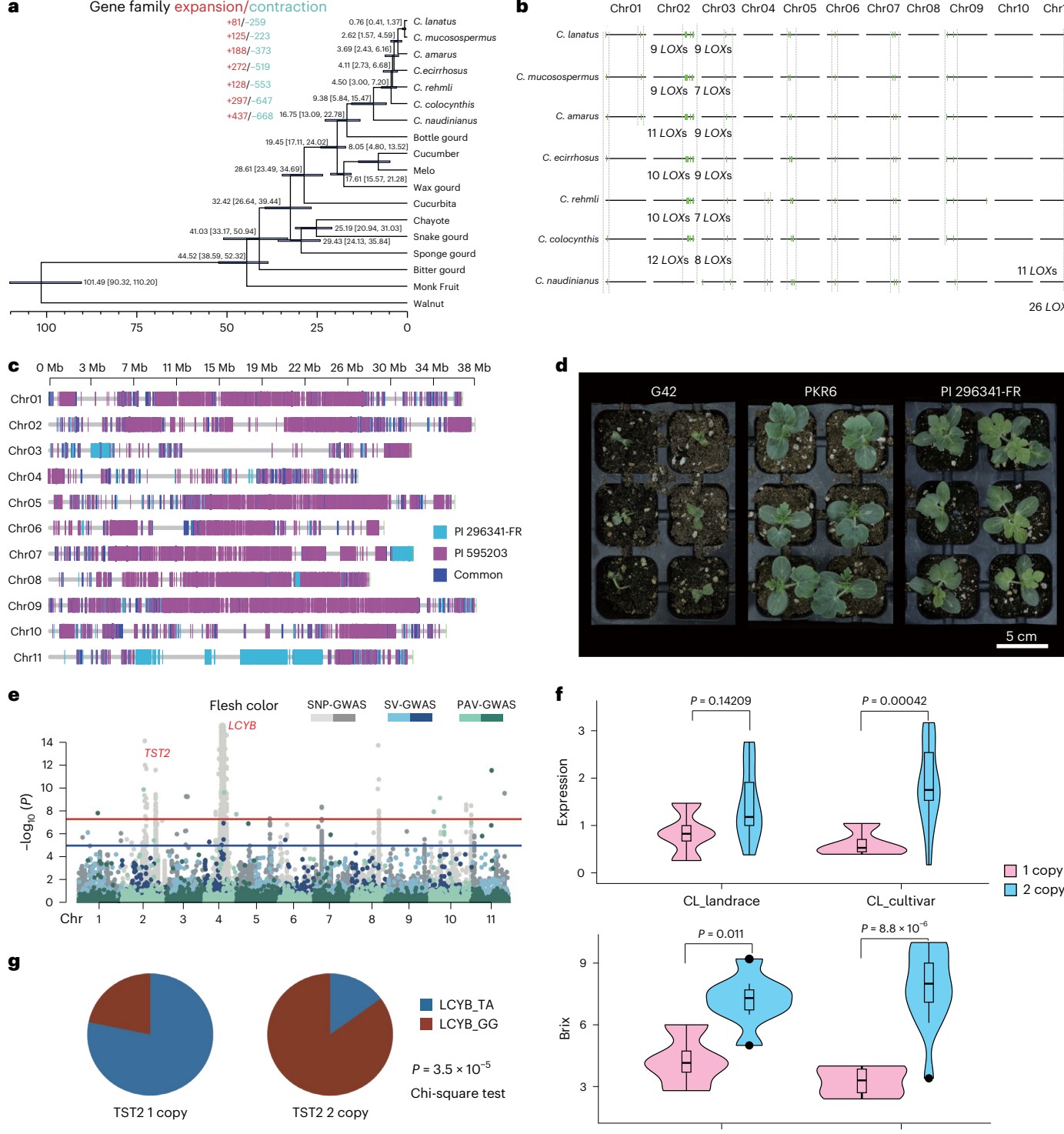

**Fig. 5 | Gene expansion and contraction are widespread and associated with agronomic trait variation during watermelon evolution. a**, Phylogenetic relationships and estimated divergence times between watermelon and other *Cucurbitaceae* species. **b**, Distribution of the candidate disease-resistant *LOX* genes across the genomes of seven watermelon species. Each green line represents an individual *LOX* gene. **c**, Graphic of genotypes of PKR6. Light blue regions indicate homozygous segments from PI 296341-FR, purple regions indicate homozygous segments from PI 595203 and dark blue represents areas of concurrence with both accessions. **d**, Phenotypes of seedlings of Fusarium wilt race 2 susceptible (G42) and resistant (PI 296341-FR and PKR6) accessions

at 14 days after inoculation. **e**, GWAS of watermelon flesh color. The *P* values corresponding to the *TST2* gene and *LCYB* gene were $7.83 \times 10^{-15}$ and $1.22 \times 10^{-16}$, respectively. **f**, Comparison of brix and *TST2* expression between accessions with CNV of *TST2*. **g**, The correlation between *TST2* copy number and genotype of *LCYB* at two SNP sites by genotyping 40 accessions. Sugar content and relative expression of *TST2* in **g** were determined by brix spindle and RT-qPCR ($n = 3$), respectively. The *P* value was determined by two-sided Wilcoxon test (**f**) or chi-square test (**g**). Data in **f** are displayed as violin plots with inner box and whisker plots. The edges and the centerlines of the boxes represent the IQR and medians, with the whiskers extending to the most extreme points (1.5× IQR).

Large inversions within the cultivated watermelon genome and its two probable ancestors are shown in a genome-based map (Fig. 4e). Inversions detected in *C. lanatus* were inherited from both *C. lanatus* subsp. *cordophanus* and *C. mucosospermus* (Supplementary Table 20). Some SVs identified in *C. lanatus* were inherited not only from *C. lanatus* subsp. *cordophanus* but also from *C. mucosospermus* (Supplementary Figs. 10 and 11), suggesting the presence of additional ancestors beyond *cordophanus* in the lineage of cultivated watermelon.

### Gene gain and loss during watermelon domestication

During watermelon domestication, gene gain and loss were instrumental in diversifying genes related to disease resistance, sugar accumulation and fruit flesh coloration[6,29]. Comparative analyses using single-copy orthologous genes across various *Citrullus* and related cucurbit species unveiled their divergence times, with *C. naudinianus* diverging approximately 16.75 million years ago (Mya) and the most recent divergence event occurring between *C. lanatus* and *C. mucosospermus* around 0.76 Mya (Fig. 5a). Notably, more genes were lost than gained during watermelon evolution and domestication (Fig. 5a), elucidating several gene changes throughout the domestication process.

PKR6 is an inbred line with multiple disease resistance pyramided from interspecific crossing, the graphic genome information of PKR6 is presented in Fig. 5c. One QTL designated *Qfon1.1* conferring resistance to Fon race 1 was identified in a 5-cM region of chromosome 1 (ref. 16). We have narrowed down the QTL to a small region (364 kb) by aligning the PKR6 genome sequence of the QTL region to the genome of G42, a susceptible line. The high-level resistance to Fon race 2 in PKR6 (Fig. 5d) makes it a better choice to use than CWRs for introgressing resistance to elite watermelons. Large linkage drags (2–5 Mb) from PI 296341-FR (*C. amarus*) were detected on Chr11 (Supplementary Table 15). The development of PKR6 demonstrates that the lost resistant genes can be integrated into elite lines on purpose with the current genomic information. Future efforts may be focused on the introgression of resistant genes from *C. naudinianus*, *C. ecirrhosus* and *C. rehmii* into elite watermelon lines.

The most substantial gains in cultivated watermelon were observed in sweetness and flesh color. Copy number variation (CNV) analysis revealed the amplification of the *TST2* genes (*ClG42_02g0107100*, *ClG42_02g0107200*) contributing to sugar accumulation and the base change of the *LCYB* gene (*ClG42_04g0042900*) leading to the coloration of watermelon flesh (Fig. 5e,f). The analysis also indicated a strong correlation ($P = 3.5 \times 10^{-5}$) between the duplicate copies of *TST2* with *LCYB*-GG and the single copy of *TST2* with *LCYB*-TA (Supplementary Table 23 and Fig. 5g), suggesting simultaneous domestication of *TST2* and *LCYB* during watermelon domestication. Moreover, high levels of expression of the chromoplast phosphate transporter *ClPHT4;2* (*ClG42_10g0214700*) were found necessary for flesh coloration, with a distinct model of sugar and phytohormone signaling mediating *ClPHT4;2* transcription in cultivated watermelons compared to wild accessions[30]. This indicates that the copy number of the *TST2* gene may serve as a threshold switch regulating the sugar signaling pathway and thus the expression of *ClPHT4;2*, leading to the coselection of watermelon sugar accumulation and fruit coloration. Reintroducing lost resistance genes and understanding the co-evolution of sugar accumulation and flesh coloration hold crucial importance for effective watermelon breeding programs.

### Contribution of SV genes during evolution and domestication

It has been reported that selection has affected a wide range of agronomic traits in watermelon fruit, including bitterness, sugar content, flesh color, shape, ripening and seed size[6]. To investigate whether the identified genes associated with these traits were influenced by environmental and/or human selection, SVs located in the promoter and CDS of *C. colocynthis*, *C. amarus*, *C. mucosospermus* and *C. lanatus* were scanned. Selective sweeps were also analyzed by comparing wild and cultivated watermelons to explore new candidate genes harboring SVs that may contribute to trait diversity (Supplementary Note 6).

We discovered that SVs of functional genes were present in *C. colocynthis* and *C. amarus* but were absent in *C. mucosospermus* and *C. lanatus*, indicating that those SVs have likely been selected against during speciation and domestication (Supplementary Table 29). SVs in functional genes were usually located upstream of the start codon, and a few were located in the coding region (Fig. 6). As reported by a study discussed in ref. 13, we noticed that gene expression could be affected by SVs, which then leads to changes in one or more agronomic traits (Fig. 6). Furthermore, SVs of genes related to fruit sweetness, bitterness and flesh color were consistent with the expression patterns in fruit of different accessions (Fig. 6 and Supplementary Fig. 13a–c).

A critical domestication trait of watermelon fruit is the loss of bitterness[31]. As noted earlier, the principal bitter compound isolated from watermelon is CuE, which has been reported to be found in the roots and fruits of wild species but only in the roots of cultivated species[32]. In *Citrullus* species, the synthesis of CuE is differentially controlled via tissue-specific regulators. Although *ClBt* (*ClG42_01g0033300*) was reported to be a fruit-specific cucurbitacin regulator and could initiate the first step of the CuE biosynthetic pathway in the fruit of watermelon[32], it seems to be expressed higher in leaf than fruit, with nearly zero expression in fruit samples of most accessions (Supplementary Table 30). In addition, a 6-bp and an 18-bp deletion located in 941 bp and 247 bp upstream of the start codon of *ClBt* were identified in *C. colocynthis* and *C. amarus*, respectively (Fig. 6). According to ref. 32, *Cl890B* (*ClG42_01g0153000*), *Cl170* (*ClG42_06g0015500*), *Cl180* (*ClG42_06g0015600*) and *ClACT* (*ClG42_06g0015800*) are also involved in the CuE biosynthetic pathway, and *ClMATE1* (*ClG42_01g0153300*)[33] is related to the transport of CuE in watermelon. SVs were detected in the promoter or CDS region of *C. colocynthis* and, to a lesser extent, in *C. amarus*. The expression levels of these five bitterness genes were highest in *C. colocynthis*, decreased sharply in *C. amarus* and were barely expressed in *C. mucosospermus* and *C. lanatus*. Expression levels were positively correlated with the CuE content (Fig. 6 and Supplementary Fig. 13a).

In contrast to reduced cucurbitacin content, sugar and pigment accumulated in the flesh of cultivated watermelon[6]. The CNV of *TST2* (ref. 7) and SVs of *ClSWEET3* (*ClG42_01g0006000*)[34], *ClVST1* (*ClG42_02g0044600*)[35], *ClNAC68* (*ClG42_03g0079100*)[18] and *ClAGA2* (*ClG42_04g0035700*)[36] were detected in the distantly related species *C. colocynthis* and *C. amarus*, the less distantly related *C. mucosospermus* and the domesticated *C. lanatus* (Fig. 6). The previously mentioned genes were less expressed in the fruit of the landraces and were barely detectable in the fruit of *C. colocynthis*, *C. amarus* and *C. mucosospermus* (Supplementary Fig. 13b). Two transcription factors, ClbZIP1 and ClbZIP2, sensed the elevated sugar content and bound to the abscisic acid (ABA)-responsive element (ABRE) motif in the promoter region of *ClPHT4;2*, thereby *ClPHT4;2* got upregulated resulting in carotenoid accumulation in cultivated watermelon species[30]. There were 6-bp and 12-bp deletions in the CDS region of *ClPHT4;2* in *C. colocynthis* and *C. amarus*, respectively (Fig. 6). A positive correlation was observed between flesh color and gene expression, as were *ClNCED1* (*ClG42_01g0254100*)[34] and *ClNCED7* (*ClG42_07g0105300*)[37] (Fig. 6 and Supplementary Fig. 13c).

Unlike bitterness, sugar content and flesh color, which are either decreasing or increasing during domestication, phenotypes of fruit shape, seed size and fruit ripening were varied between different species. During domestication, fruit shape changes from small to bigger and then slightly smaller, and seed size is also variable (Fig. 1a–c). Although the gene expression patterns have been reported to be associated with fruit shape and seed size traits[19,38,39], they do not coincide with the appearance of SV (Fig. 6 and Supplementary Fig. 13e). SVs of two genes related to fruit ripening existed in *C. colocynthis* and *C. amarus*, but they are inconsistent with the expression among the four species

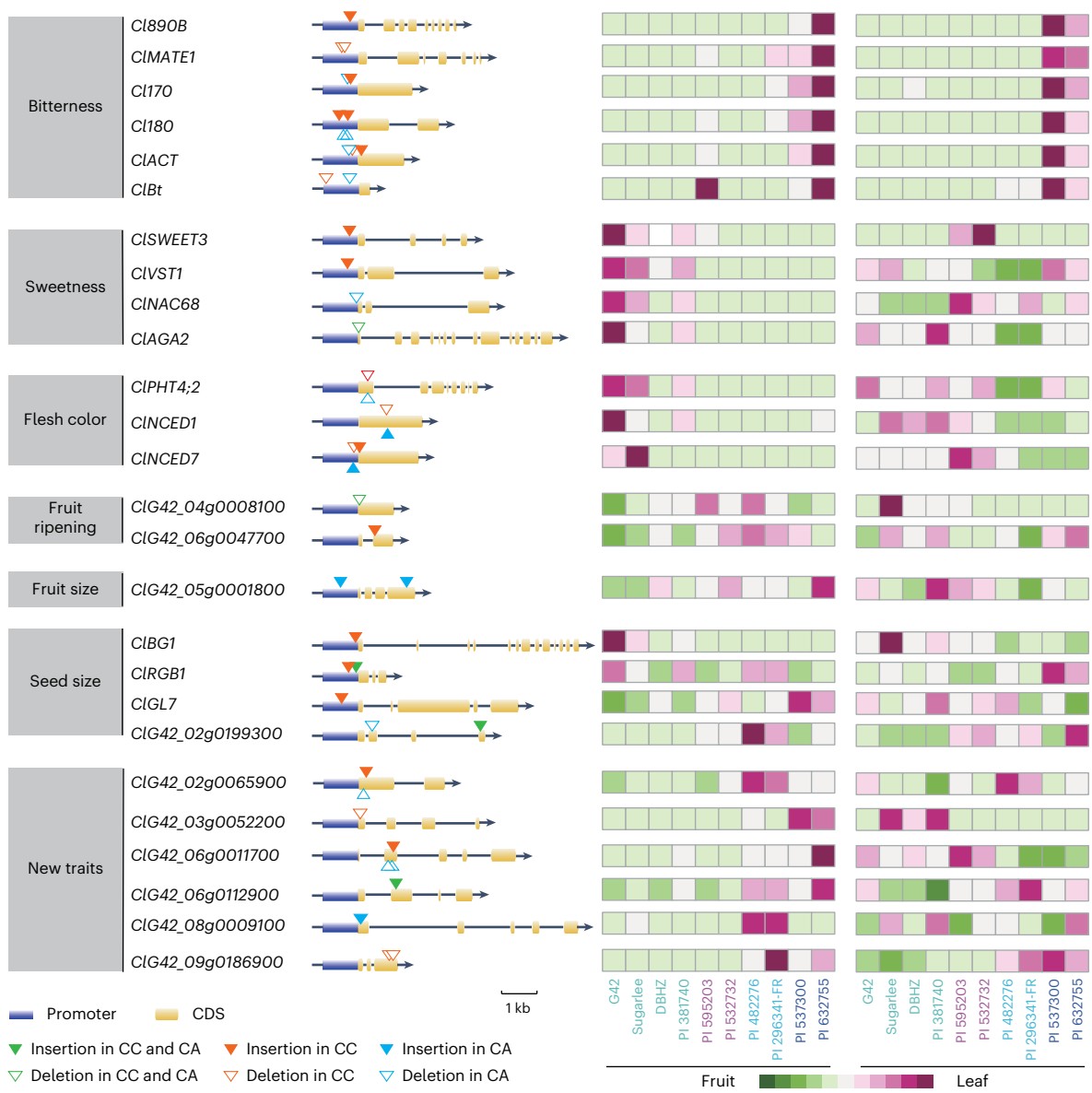

**Fig. 6 | SV distributions and expression levels of genes that relate to agronomic traits in wild and cultivated watermelon accessions.** The overall performance of using SV and differential expression to predict associated genes is shown (Methods). Triangles marked with different colors represent the SVs that existed in *C. colocynthis* and/or *C. amarus* listed in Supplementary Table 1. The gray squares on the left represent the agronomic traits and selective sweeps. Agronomic-traits-related genes were previously reported, while predicted genes from selective sweeps are newly identified by homologous comparison. The tissues used for expression profiling are indicated at the bottom of each column.

(Fig. 6 and Supplementary Fig. 13d). The process of domestication is complicated. SV is just one of the factors affecting variations among *Citrullus* species.

## Discussion

We have publicly released a T2T genus-level super-pangenome, which consists of 27 T2T gap-free genomes that include all seven watermelon (*Citrullus*) species. Among them, the genomes of *C. rehmii*, *C. ecirrhosus* and *C. naudinianus* species have not previously been available. Our results suggest that the use of species-specific references in sequence alignments improves alignment quality. The sequencing of telomeres and centromeres in all 27 watermelon accessions revealed enrichment for LTR (Gypsy) elements in the centromere region. The representative monomer in cluster 1 obtained by phylogenetic tree clustering through 27 watermelon accessions monomers existed widely in watermelon germplasm (Supplementary Table 13). Three major chromosomal

rearrangements (Supplementary Table 31) that occurred during *Citrullus* evolution were identified and validated. These rearrangements likely influence the expression and duplication of hundreds of genes. The comparison of SVs present in *C. lanatus cordophanus*, *C. mucosospermus* and the cultivated watermelon (*C. lanatus*) provides a more fully developed picture to help us understand the relationship of the cultivated watermelon with its two nearest relatives. Our findings indicate that both *C. lanatus* subsp. *cordophanus* and *C. mucosospermus* made unique SV contributions to the *C. lanatus* genome.

PKR6 is a recombinant inbred line derived from *C. lanatus*, *C. mucosospermus* and *C. amarus*, created to study and exploit the genetic bases of multiple disease resistances. By aligning the published intervals of disease resistance QTLs named *Qfon1.1* (ref. 16), we were able to narrow the intervals and increase the likelihood of detecting disease resistance genes (*ClG42_01g0002300*, *ClG42_01g0002600* and *ClG42_01g000440*). Our program has produced inbred lines derived from *C. colocynthis* and

*C. ecirrhosus* (Fig. 1a,b) using the conventional crossing and backcrossing methods, but at great costs in time and money. However, the watermelon T2T genus-level super-pangenome facilitates a better understanding of where genes of interest lie in the context of the structural rearrangements in the genomes of the cultivated and wild species.

The characterization of SVs in the *Citrullus* species genomes enabled a functional investigation during watermelon domestication and breeding by population analysis. Presence/absence variation genome-wide association study (PAV-GWAS) identified *TST2* and *LCYB* that were correlated to sugar content and flesh color. The copy number of *TST2* was associated with a single allele of *LYCB*, indicating that fruit flesh sweetness and red flesh coloration were selected simultaneously. The comprehensive SV landscape described in this report highlights the narrow genetic diversity of cultivated watermelon and the remarkable abundance of genetic variation in its CWRs. Despite the wild relatives involve mechanisms to cope with a wide range of biotic and abiotic stresses, these genetic resources are rarely used in breeding programs, although all six CWRs can be hybridized with *C. lanatus*[6]. The T2T genus-level watermelon super-pangenome provided in this study should help to better understand the advantages and disadvantages of hybridization by providing a basis for predicting introgression-related problems. We acknowledge that references generated from the 27 samples are insufficient to capture all the sequence diversity present at the population and species levels, especially for *C. ecirrhosus*, *C. rehmii* and *C. naudinianus* because the availability of germplasm for these CWRs is extremely limited. A more diverse reference map will surely expand our understanding of the genomics and genetic diversity of watermelon and its gene pools.

## Online content

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

## Methods

### Sample selection and sequencing

The 27 representative accessions covering all species of watermelon were collected throughout the world, including 1 *C. naudinianus*, 4 *C. colocynthis*, 1 *C. rehmii*, 1 *C. ecirrhosus*, 5 *C. amarus*, 2 *C. mucosospermus* (Egusi) and 13 *C. lanatus* (5 *C. lanatus* landrace and 8 *C. lanatus* cultivar; Supplementary Table 1). *C. lanatus* subsp. *cordophanus* (*cordophanus*; PI 254622) originated from Sudan. All accessions were grown in the greenhouse at Weifang, Shandong, China (119°18′E, 36°16′N).

For genome sequencing, leaf samples were collected from all accessions at the vegetative growth stage. For RNA-seq and metabolite analysis, fruit flesh samples of 11 accessions, including 2 *C. colocynthis*, 1 *C. rehmii*, 2 *C. amarus*, 2 *C. mucosospermus*, 2 *C. lanatus* landrace and *C. lanatus* cultivar, were collected from five different developmental stages (10, 18, 26, 34 and 42 days) after pollination. All samples were snap-frozen in liquid nitrogen (LN) and sent in dry ice to companies for sequencing.

### Plant phenotyping and evaluation of agronomic traits

Twenty-seven accessions were grown in the greenhouse. The agronomic phenotypes, including leaf, flowers, fruit flesh color, soluble solid content and seeds (Fig. 1 and Supplementary Fig. 1), were recorded during the watermelon growing stages. To phenotype the Fusarium wilt resistance of the PKR6 (*C. lanatus*) accession, two true leaf stage seedlings were inoculated with conidial suspension of *Fusarium oxysporum* f. sp. *niveum* race 2 at a concentration of $1 \times 10^6$ ml$^{-1}$, and watermelon cultivar G42 and the wild watermelon PI 296341-FR (*C. amarus*) were used as the susceptible and resistant control.

### Preparation of short Illumina libraries and data processing

Portions of the DNA were sent to BerryGenomics to construct Illumina DNA libraries and for sequencing using the Illumina Hiseq platform. The short reads were filtered. This involved the elimination of adaptor contamination and low-quality reads by Fastp[40] with default parameters. Short reads generated from the Illumina platform were used for the estimation of genome size, the level of heterozygosity, mapping rate and genome coverages.

### HiFi library preparation and sequencing

The extracted DNA molecules were sequenced using both the Illumina and PacBio Sequel (Pacific Biosciences of California) platforms. Portions of the DNA were sent to BerryGenomics to construct circular consensus sequencing libraries containing about 15 kb of cut fragments and sequenced using a PacBio Sequal platform. The HiFi reads from the PacBio platform were used for genome assembly.

### Hi-C library construction and sequencing

High-grade DNA extracted from young leaves was used for Hi-C sequencing. Chromatin was stabilized using formaldehyde, and standard in situ Hi-C chromosome conformation capture was carried out[41]. Libraries were sequenced using a 150 bp paired-end mode on an Illumina NovaSeq platform (Illumina).

### Nanopore ultra-long library construction and sequencing

For ultra-long sequencing using Oxford Nanopore, libraries were prepared per the standard protocol and using the SQK-LSK110 ligation kit. The purified library was subsequently loaded onto a primed R9.4.1 Spot-On Flow Cell and sequenced on a PromethION sequencer. The ultra-long reads from the ONT platform were used for genome assembly and filling gaps.

### BioNano data generation and construction of optical maps

Using the Saphyr Genome Imaging Instrument and the DNA ligase 1 (DLE1) non-nicking enzyme (CTTAAG), bionano optical maps were prepared for the investigation of genomic structure.

High-molecular-weight DNA was extracted, nicked and labeled using the SaphyrPrep Kit. The labeled DNA was subsequently imaged using the Saphyr system. The imaged molecules were assembled via Bionano Solve software (v.3.3). The assembled genomic alignments were visually represented using Bionano Access software (v.1.5.2, https://bionanogenomics.com/support/software-downloads/).

### Metabolite profiling

Freeze-dried watermelon fruit tissue was ground into a fine powder using a mixer mill for 40 s at 60 Hz. In total, 50 mg of sample powder was mixed with 1 ml of cold (4 °C) 70% (vol/vol) aqueous methanol and extracted overnight at 4 °C. The extracts were then sedimented by centrifugation at 12,700 rpm at 4 °C for 20 min. The supernatants were passed through a 0.22 μm millipore filter and used for the following liquid chromatography–mass spectrometry analysis. A quality control sample was also prepared by pooling aliquots of sample extracts.

### Genome survey

The Illumina short reads were used for survey analysis, and the *k*-mer distributions were estimated using Jellyfish[42] with parameters -m 21 -t 1 -s 5G -C. Then the genome sizes and heterozygosity rate were calculated by the web tool Genomescope1.0 (ref. 43).

### T2T genome assembly of the 27 watermelon accessions

For the T2T genome assembly of the 27 watermelon accessions, the four steps were followed. Step 1—the following five tools, each based on distinct algorithms, were used to assemble the watermelon genomes: (i) Hifiasm (v0.16.1) was used with default parameters for assembly from HiFi reads[44]; (ii) Hifiasm (v0.18.4) was used with the '--ul' parameter to assemble genomes from both HiFi and ultra-long reads[45]; (iii) NextDenovo[46] was used for preliminary genome assembly using ultra-long reads, with the following parameters: genome_size = 380 m, read_type = ont, rerun = 3 and read_cutoff = 50,000; (iv) Hifiasm (v0.16.1) was again used to assemble the genomes from HiFi reads, this time using the parameters '--write-ec --write-paf -l0'. The draft contigs were subsequently assembled into a gapless genome by PGA (https://github.com/likui345/PGA).[47] (v) The La Jolla Assembler was used with default parameters for genome assembly from HiFi reads[48].

Step 2—HiC-Pro[49] was used to categorize Hi-C data into valid and invalid interaction pairs, retaining only the valid pairs for further assembly. EndHiC[50] was used to assemble large contigs into chromosomal-level scaffolds, using Hi-C links originating from contig end regions as opposed to entire contig regions. Subsequently, a heatmap illustrating genomic interactions was constructed using HiCPlotter software[51].

Step 3—based on the results of these five software tools, the Ragtag patch was then used to integrate and make fill-gaps in the target genome using sequences from the query genome[52].

Step 4—to identify telomeres, the telomere pipeline developed by the Vertebrate Genome Project (https://github.com/VGP/vgp-assembly) was used and identified plant telomeric sequences (CCCTAAA)[53]. We performed manual identification of telomeric reads (HiFi and ONT reads), using the minimap2 (ref. 54) alignment tool to map those reads and incorporate the corresponding telomeric sequences to patch any telomere-deficient chromosomes. The genome assembly was refined through six iterative rounds of polishing using both the Racon and Merfin algorithms[55].

### Identification of centromere and telomere sequence

TelomereSearch (https://github.com/jamiemcg/TelomereSearch) was used to identify the telomere range.

A pipeline comprising the TRF tool[56] and CD-HIT[57] tool was used to identify the centromere region. The TRF tool is used to detect tandem repeats and monomers throughout the genome. CD-HIT was used to cluster these monomers, reducing sequence redundancy and enhancing the accuracy of centromere localization through sequence similarity

search. By using the centromere region identified in watermelon G42 as a reference, the continuous and high-frequency regions are identified as potential candidate centromere regions. Finally, the results of gene density and Gypsy LTR numbers are integrated with the candidate regions to predict the most probable location of the centromere. The top eight monomer clusters of each watermelon variety were counted, and the monomer clusters were compared by MAFFT[58]. The maximum likelihood method in FastTree is used to construct the evolutionary tree[59].

### Genome evaluation
Genome completeness was evaluated by BUSCO using the embryophyta_odb10 database[60]. Genome continuity was evaluated by calculating the contig N50 length and mapping coverages by qualimap2. The accuracy of the genome is evaluated by calculating the sequencing data mapping rate and quality value (QV)[61]. Finally, the LTR assembly index (LAI) value, which uses repeat sequences, evaluated the assembly level of the genome[62,63].

### Genome annotation
**TE sequence annotation.** Annotation of the genome repeat sequences was performed using Extensive de novo TE Annotator software[64]. The first step involved conducting de novo identification of repetitive elements, where individual TE libraries were constructed with parameters '-species others --sensitive 1 --anno 1 --evaluate 1'. These libraries served as crucial inputs for the subsequent step of structural annotation of the TEs. RepeatMasker was used to mask the genome and annotate the TE elements with a high-quality nonredundant TE library with parameters '-e rmblast -s -gff -nolow -no_is -norna'[65].

### Protein-coding gene structure prediction
The structure of the protein-coding genes was predicted using the following three methods: ab initio gene prediction[66–68], homology-based gene prediction[6,69–72] and RNA-seq data gene prediction[73–75]. EVidenceModeler was used to integrate all the prediction results of the three methods to predict gene models[76]. Finally, gene models were filtered by removing the gene coding sequences that overlapped with the TE sequences by more than 20% or when the coding region length was less than 200 bp.

### Noncoding RNA gene annotation
tRNAscan-SE[77] was used to identify tRNA genes with default parameters. RNAmmer[78] was used to predict rRNA sequences, and Infernal[79] was used to search snRNAs from the Rfam database.

### Gene function annotation
Four approaches were used to predict the function of protein-coding genes. The first was to use BlastP to search against protein sequences in the National Center for Biotechnology Information (NCBI) nonredundant protein database[80] and the Swiss-Prot database[71]. Second, protein domain and gene ontology term annotations were performed using InterProScan[81]. The third was the Kyoto Encyclopedia of Genes and Genomes (KEGG) annotation with KEGG Automatic Annotation Server[82]. The fourth was eggNOG-mapper annotation with an online resource server[83].

### Organellar genome assembly and annotation
For the assembly of mitochondrial and chloroplast genomes, GetOrganelle software[84] was used. This software is specifically designed for the de novo assembly of organelle genomes from whole-genome sequencing data. For the chloroplast genome assembly, we applied parameters '-F embplant_pt R 10 -t 52 -k 21,45,65,85,105', while for the mitochondrial genome, parameters '-F embplant_mt -R 30 -k 45,65,85,105,115' were used.

### Phylogenetic tree analyses of 429 watermelon resequencing
SNP calling was performed on 27 selected watermelon samples in this study, following the methodology of a study discussed in ref. 6

Subsequently, we merged the variations (VCF) of these 27 samples with the common variations of 402 samples using bcftools[85] merge, which facilitated the construction of a phylogenetic tree. The phylogenetic tree Newick file was generated using VCF2Dis software. To enhance the visual representation of the tree, it was further refined using Interactive Tree Of Life software[86].

### Gene family identification and phylogenetic analysis
For the gene family analysis, the longest predicted protein from each individual gene was used as the input and processed by OrthoFinder[87] with the parameters '-S diamond -M msa -T fasttree'. The output from this procedure includes comprehensive statistical information, and a species tree is constructed using single-copy genes.

### Core and dispensable gene family of pangenome construction
Gene families shared among accessions or species were characterized as core gene families. Gene families absent in one or two accessions were classified as softcore gene families, whereas gene families missing in more than two accessions or one species were identified as dispensable gene families. Furthermore, gene families exclusive to one accession or species were recognized as private gene families. Both core genome and pangenome size estimates underwent a series of random samplings with 100 repetitions for each accession count. For the phylogenetic assessment of each gene family, we used MUSCLE[88] (v3.8.31) to perform sequence alignment and MEGA7 (ref. 89) to construct the phylogenetic tree and compute the nucleotide diversity ($\pi$ value).

### Whole-genome alignment and SV identification
Genome assemblies were harmonized with the G42 reference genome using MUMmer4, using parameters '-c 100 -b 500 -l 50 –maxmatch'. Subsequent filtration was carried out using delta filter with parameters '-m -i 90 -l 100' as per the methodology outlined in ref. 90 We used MCscan to generate both intragenomic and intergenomic homologous and collinear blocks among seven watermelon species[91]. This facilitated the analysis of chromosome recombination across different species. These alignments were then used for variation detection via the SyRI pipeline[92]. Identification of SNPs and InDels among all watermelon materials, in comparison to G42, was achieved through the use of Anchorwave software[93].

### Validation of chromosome fission and fusion
We picked one accession of each species (except for *C. mucosospermus*) to validate chromosomal translocations using the PCR method. We selected regions with high identities when compared to the reference genome and designed two sets of primers (Supplementary Table 14) in the translocated regions. Each set contained a forward primer specific to the G42 genome upstream of the breakpoint and forward and reverse primers inside the translocated region. Two forward primers combined with the reverse primer, respectively, were used to perform PCR.

### A and B compartment identification and analysis
To identify A and B compartments, we adopted a method based on principal component analysis (PCA). In essence, we assigned compartments A and B according to the Hi-C interaction matrix along with gene and TE densities. The HiC-Pro was used to obtain normalized and iterative correction and eigenvector decomposition (ICE) interaction matrices, calculated at a resolution of 50 kb for each chromosome[49]. Following this, we computed the Pearson correlation matrix and the covariance matrix on the observed/expected matrices. Ultimately, the R package HiTC was used to calculate PCA eigenvectors on the covariance matrix, leading to the identification of compartments A and B[94].

### Gene family analysis and estimation of divergence times
Clustering of orthologous gene groups was performed among the *Cucurbitaceae* species using OrthoFinder, using its default parameters. For the construction of the species tree, conserved single-copy genes

were used. The CAFE5 tool was used to detect expansions and contractions within the gene families, again using default parameters[95]. The MCMCtree program facilitated the estimation of divergence times[96]. Using the TimeTree database, we selected the divergence times of walnut and monk fruit as calibration points[97].

## Annotation of resistance genes

Disease resistance genes in watermelon were identified by using the Plant Resistance Genes database (PRGdb4.0)[98]. The Disease Resistance Analysis and Gene Orthology (DRAGO 3) tool was used as a pipeline for annotating these resistance genes[99]. To identify lipoxygenase (LOX) proteins, watermelon protein sequences were compared against the InterPro database[100], specifically searching for the presence of the lipoxygenase domain (either IPR001024 or IPR000907).

## SV/PAV genotyping and SV/PAV genome-wide association studies

Before GWAS, stringent filtering selected 9,618,384 SNPs meeting criteria like minor allele frequency (MAF) ≥ 0.01 and ≤50% missing data. The mixed linear model drove the GWAS, setting a significant SNP threshold at $5.19 \times 10^{-9}$ via Bonferroni correction. GWAS results were depicted using the R package qqman (https://github.com/stephenturner/qqman).

BayesTyper[101] mined SV mutations, leaving 138,124 SVs after remaining MAF ≥ 0.01 and ≤50% missing data variations. BWA-MEM (https://github.com/lh3/bwa) compared the watermelon population's second-generation sequencing data with >50 bp PAVs from G42. Filtering and sorting yielded 157,060 variations in VCF format. GWAS analyzed SVs and PAVs like SNPs.

## Genotyping of 42 accessions at *TST2* and *LCYB* loci

Forty-two accessions from different species were used to genotype *TST2* and *LCYB* loci (Supplementary Table 23). CNV of *TST2* was detected using the method as reported[7]. For the *LCYB* locus, we sequenced the PCR product of *LCYB* using Sanger sequencing technology. Two reported SNPs of *LCYB*[102] were used to genotype 42 accessions.

## Integrated transcriptomic and metabolomic analysis

PCA was performed on the high-variance genes and metabolite content across five fruit developmental stages from 11 watermelon samples. The expression matrix of 3,078 detected metabolites was log-transformed for weighted gene co-expression network analysis[103]. Differential metabolites were assessed based on Pearson's correlation method to calculate the relationship between all genes and metabolites. Following this, genes and metabolites that exhibited high expression correlation based on the correlation coefficient and the *P* value of the correlation were identified.

## RNA extraction and quantitative real-time PCR analysis

Total RNA was extracted using TransZol Up (TransGen Biotech) following the protocol provided. The first-strand cDNA was synthesized using the HiScript III RT SuperMix for qPCR (+gDNA wiper; Vazyme). qPCR was performed on a QuantStudio (Thermo Fisher Scientific) instrument using ChamQ SYBR qPCR Master Mix (Vazyme). Actin was taken as an internal reference.

## SV validation

**SV between cultivar and wild species.** Twenty-four accessions listed in Supplementary Table 1 except for *C. naudinianus*, *C. ecirrhosus* and *C. rehmii* were used to validate SVs (200 bp to 10 kb) detected using specific primers. The primer sequences are summarized in Supplementary Table 14, and electrophoresis results are given in Supplementary Fig. 3.

## PCR validation of inversions

Accessions of *C. lanatus* and *C. mucosospermus* listed in Supplementary Table 1 were used to validate eight identified inversions. One pair of primers was designed for small fragments of inversions, while two pairs

of primers were designed close to both ends named upper and lower, respectively. PCR products were sequenced using Sanger sequencing technology. Sequence chromatograms were analyzed with SnapGene software, and the results are shown in Supplementary Fig. 5.

## SV in candidate genes

To validate SV in candidate genes associated with fruit shape, flesh color and fruit flavor, DNA samples from accessions of *C. colocynthis*, *C. amarus* and *C. lanatus* were used as templates. Specific primers (Supplementary Table 18) were designed for each candidate gene. The PCR products were sequenced by Sanger sequencing technology (Tsingke Biotech). Sequence chromatograms were analyzed with SnapGene software.

## qRT–PCR analysis of candidate genes

To evaluate the transcript levels of candidate genes listed in Fig. 6, we designed qRT–PCR primers by Primer3Plus. The expression of the candidate gene was investigated using the watermelon actin-7 (*ClG42_02G0007100*) gene to calibrate its expression level as previously described[18]. The $2^{-\Delta\Delta Ct}$ method was used to quantify relative gene expression levels[104,105]. RNA samples of fruit flesh were used as a template. The sequences of all primers used in this study are listed in Supplementary Table 32. qPCR was performed according to the protocol of the ChamQ SYBR qPCR Master Mix (Vazyme).

## Graph-based genome construction

Using the genome of 26 watermelon accessions reassembled in this study, a graphical pangenome was created via minigraph[106]. The resulting pangenome was converted using VG Software (https://github.com/vgteam/vg) and mapped with BWA-MEM (https://github.com/lh3/bwa). The alignment rates between this study and previously published watermelon varieties were summarized using SAMtools[107].

## Statistical analysis

The statistical analyses were performed in R (v4.0.2). A two-sided Wilcoxon test was used to compare the difference in gene expression levels and sugar content between two groups of samples. A two-sided chi-squared test was used to determine the statistical difference in the correlation between *TST2* copy number and *LCYB* genotype. One-way analysis of variance with Tukey's test was used to determine the difference in relative expression of 27 candidate genes among different species. A two-sided Student's *t* test was used to compare the difference in gene expression levels and metabolite contents.

## Reporting summary

Further information on research design is available in the Nature Portfolio Reporting Summary linked to this article.

## Data availability

The raw sequencing data and genome assemblies (CP155009-CP155019 and CP155142-CP155416) have been deposited in the NCBI under the Bioproject (PRJNA1031825) and the National Genomics Data Center (NGDC, https://ngdc.cncb.ac.cn/) under the Bioproject (PRJCA020693). The genome assembly, annotations and graph pangenome are also available in the Watermelon Genome Database (http://www.watermelondb.cn). Source data are provided with this paper.

## Code availability

The code used for this paper is available on GitHub (https://github.com/yilinZhang-bio/Watermelon-pangenome) and Zenodo[108] (https://doi.org/10.5281/zenodo.10693455).

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

## Acknowledgements

This work was supported by the Key R&D Program of Shandong Province, China (ZR202211070163), the Provincial Technology Innovation Program of Shandong, the Ningxia Hui Autonomous Region Agricultural Breeding Special Project (NXNYYZ202001), the Ningbo Science and Technology Innovation Project (2021Z132) and the Weifang Seed Innovation Group. We would like to thank the National Mid-Term Genebank for Watermelon and the Melon and Chinese Academy of Agricultural Sciences for providing wild and cultivated watermelon resources. We would like to thank Y. Xu from the National Engineering Research Center for Vegetables and the Beijing Academy of Agriculture and Forestry Sciences for providing seeds for some watermelon accessions. We would also like to thank J. Leebens-Mack of the Department of Plant Biology at the University of Georgia for his valuable help in revising this manuscript.

## Author contributions

X.Z., H.H., X.W.D. and Y.D. conceived the idea and designed the project. H.H. coordinated the entire bioinformatic analysis. X.Z., Y.D., X.C., M.Z., Y.T. and Y.S. collected and propagated the samples and investigated phenotypes. L.X. and X.L. performed disease evaluation and phenotyping. Y.Z. performed all bioinformatics analyses. M.H., Y.Z. and X.H. performed centromere and telomere analyses. D.L. and B.L. performed adaptive sequencing. Y.Z. and Y.C. constructed the genome database. M.H. and Y.Z. conducted the GWAS analyses. Y.Z., M.Z., J.T. and Y.D. completed the SV and gene function analysis. Y.Z. and X.H. contributed to the chloroplast and mitochondrial genome assembly. M.Z., J.T. and T.F. did the molecular biology experiments. Y.L., G.Q. and Y.Z. performed metabolome experiments and analyses. Y.Z., M.Z., M.H., J.T., H.H., X.Z., X.W.D. and Y.D. coordinated to all co-authors and wrote the manuscript. R.J. provided plant materials and suggestions for the project and edited the manuscript.

## Competing interests

The authors declare no competing interests.

## Additional information

**Correspondence and requests for materials** should be addressed to Yun Deng, Xingping Zhang or Hang He.

| | |
|---|---|

# Reporting Summary

## Statistics

For all statistical analyses, confirm that the following items are present in the figure legend, table legend, main text, or Methods section.

| n/a | Confirmed | |
|---|---|---|
| ☐ | ☒ | The exact sample size (*n*) for each experimental group/condition, given as a discrete number and unit of measurement |
| ☐ | ☒ | A statement on whether measurements were taken from distinct samples or whether the same sample was measured repeatedly |
| ☐ | ☒ | The statistical test(s) used AND whether they are one- or two-sided<br>*Only common tests should be described solely by name; describe more complex techniques in the Methods section.* |
| ☒ | ☐ | A description of all covariates tested |
| ☐ | ☒ | A description of any assumptions or corrections, such as tests of normality and adjustment for multiple comparisons |
| ☐ | ☒ | A full description of the statistical parameters including central tendency (e.g. means) or other basic estimates (e.g. regression coefficient) AND variation (e.g. standard deviation) or associated estimates of uncertainty (e.g. confidence intervals) |
| ☐ | ☒ | For null hypothesis testing, the test statistic (e.g. *F*, *t*, *r*) with confidence intervals, effect sizes, degrees of freedom and *P* value noted<br>*Give P values as exact values whenever suitable.* |
| ☒ | ☐ | For Bayesian analysis, information on the choice of priors and Markov chain Monte Carlo settings |
| ☒ | ☐ | For hierarchical and complex designs, identification of the appropriate level for tests and full reporting of outcomes |
| ☒ | ☐ | Estimates of effect sizes (e.g. Cohen's *d*, Pearson's *r*), indicating how they were calculated |

*Our web collection on statistics for biologists contains articles on many of the points above.*

## Software and code

Policy information about availability of computer code

| Data collection | Data were sequenced from PacBio, Nanopore and Illumina platform. |
|---|---|
| Data analysis | Fastp (v0.23.2; https://github.com/OpenGene/fastp); Jellyfish (v2.3.0; https://github.com/gmarcais/Jellyfish); Hifiasm (v0.18.4 &v0.16.1; https://github.com/chhylp123/hifiasm); PGA (v0.2; https://github.com/likui345/PGA); NextDenovo (v2.5.0; https://github.com/Nextomics/NextDenovo); LJA (v0.2; https://github.com/AntonBankevich/LJA); HiC-Pro (v3.1.0; https://github.com/nservant/HiC-Pro ); EndHiC (v1.0; https://github.com/fanagislab/EndHiC); HiCPlotter (v0.6.6; https://github.com/akdemirlab/HiCPlotter); RagTag (v2.1.0; https://github.com/malonge/RagTag); BUSCO (v5.4.3; https://busco.ezlab.org); RNAmmer (v1.2; https://services.healthtech.dtu.dk/software.php); InterProScan (v5,https://www.ebi.ac.uk/interpro/); BioNano Solve (v3.3; https://bionanogenomics.com/); Bionano Access (v1.5.2;https://us.bionanoaccess.com/); Samtools (v1.19; https://samtools.sourceforge.net/); Minimap2 (v2.24; https://github.com/lh3/minimap2); bcftools (v1.14; https://github.com/samtools/bcftools); iTOL (v6; https://itol.embl.de/itol.cgi); MUSCLE (v3.8.31; http://www.drive5.com/muscle/); MEGA7 (v7; https://www.megasoftware.net/); Genomescope (v1.0; https://github.com/schatzlab/genomescope); VGP (v2.0; https://github.com/VGP/vgp-assembly); TelomereSearch (v1.0; https://github.com/jamiemcg/TelomereSearch); TRF (v4.09; https://tandem.bu.edu/trf/home); CD-HIT (v4.8.1; https://github.com/weizhongli/cdhit/); MAFFT (v7.475; https://mafft.cbrc.jp/alignment/software/); FastTree (v2.1; http://meta.microbesonline.org/fasttree/); qualimap2 (v2.2.2; http://qualimap.conesalab.org); EDTA (v2.0.0; https://github.com/oushujun/EDTA); RepeatMasker (v4.1.1; https://www.repeatmasker.org); EVidenceModeler (v1.1.1; https://github.com/EVidenceModeler/EVidenceModeler); tRNAscan-SE (v2.0; http://lowelab.ucsc.edu/tRNAscan-SE/); GetOrganelle (v1.7.7.0; https://github.com/Kinggerm/GetOrganelle); VCF2Dis (v1.5; https://github.com/BGI-shenzhen/VCF2Dis); OrthoFinder (v2.5.4; https://github.com/davidemms/OrthoFinder); Mummer4 (v4.0.0; https://github.com/mummer4/mummer); SyRI (v1.6.3; https://github.com/schneebergerlab/syri); Anchorwave (v1.0.0; https://github.com/baoxingsong/anchorwave); HiC-Pro (v3.1.0; https://github.com/nservant/HiC-Pro); HiTC (v3.18; https://bioconductor.org/packages/release/bioc/html/HiTC.html); CAFE5 (v1.1; https://github.com/hahnlab/CAFE5); MCMCtree (v3.18; https://bioconductor.org/packages/release/bioc/html/HiTC.html); BayesTyper (v1.5; https://github.com/bioinformatics-centre/BayesTyper); vg (v1.45; https:// |

github.com/vgteam/vg); minigraph (v1.0; https://github.com/lh3/minigraph); Custom codes (https://github.com/yilinZhang-bio/Watermelon-pangenome)

For manuscripts utilizing custom algorithms or software that are central to the research but not yet described in published literature, software must be made available to editors and reviewers. We strongly encourage code deposition in a community repository (e.g. GitHub). See the Nature Portfolio guidelines for submitting code & software for further information.

## Data

Policy information about availability of data

All manuscripts must include a data availability statement. This statement should provide the following information, where applicable:
- Accession codes, unique identifiers, or web links for publicly available datasets
- A description of any restrictions on data availability
- For clinical datasets or third party data, please ensure that the statement adheres to our policy

The raw sequencing data have been deposited in the National Genomics Data Center (NGDC) under the Bioproject (PRJCA020693) and National Center for Biotechnology Information (NCBI) under the Bioproject (PRJNA1031825). The genome assembly, annotations and graph-pangenome are also available in Watermelon Genome Database (http://www.watermelondb.cn).

## Research involving human participants, their data, or biological material

Policy information about studies with human participants or human data. See also policy information about sex, gender (identity/presentation), and sexual orientation and race, ethnicity and racism.

| Reporting on sex and gender | N/A |
| --- | --- |
| Reporting on race, ethnicity, or other socially relevant groupings | N/A |
| Population characteristics | N/A |
| Recruitment | N/A |
| Ethics oversight | N/A |

Note that full information on the approval of the study protocol must also be provided in the manuscript.

# Field-specific reporting

Please select the one below that is the best fit for your research. If you are not sure, read the appropriate sections before making your selection.

☒ Life sciences        ☐ Behavioural & social sciences        ☐ Ecological, evolutionary & environmental sciences

For a reference copy of the document with all sections, see nature.com/documents/nr-reporting-summary-flat.pdf

# Life sciences study design

All studies must disclose on these points even when the disclosure is negative.

| Sample size | 27 accessions were used to represent species from Citrullus genus to build pan-genome. The logic of this selection was to cover the entire Citrullus genus, the number of accessions for the wild species was based on the species that are collectible. |
| --- | --- |
| Data exclusions | No data were excluded from the analysis in the article. |
| Replication | Three biological replicates with three technical replicates were used in the qRT-PCR experiment. Five biological replicates with 5 technical replicates were used in the LC-MS experiment for metabolite analysis. (Or: Replications for each experiment were clearly stated in figure legends or Methods section.) |
| Randomization | Leaves, male flowers and female flowers were randomly collected from watermelon individuals with same growth stages. The sampling process for genome DNA/RNA sequencing was randomly conducted. |
| Blinding | Blinding was not applicable to this study because no patient treatment and experiment was applied in this study. |

# Reporting for specific materials, systems and methods

We require information from authors about some types of materials, experimental systems and methods used in many studies. Here, indicate whether each material, system or method listed is relevant to your study. If you are not sure if a list item applies to your research, read the appropriate section before selecting a response.

## Materials & experimental systems

| n/a | Involved in the study |
|-----|----------------------|
| ☒ | ☐ Antibodies |
| ☒ | ☐ Eukaryotic cell lines |
| ☒ | ☐ Palaeontology and archaeology |
| ☒ | ☐ Animals and other organisms |
| ☒ | ☐ Clinical data |
| ☒ | ☐ Dual use research of concern |
| ☐ | ☒ Plants |

## Methods

| n/a | Involved in the study |
|-----|----------------------|
| ☒ | ☐ ChIP-seq |
| ☒ | ☐ Flow cytometry |
| ☒ | ☐ MRI-based neuroimaging |

# Dual use research of concern

Policy information about dual use research of concern

## Hazards

Could the accidental, deliberate or reckless misuse of agents or technologies generated in the work, or the application of information presented in the manuscript, pose a threat to:

| No | Yes |
|-----|-----|
| ☒ | ☐ Public health |
| ☒ | ☐ National security |
| ☒ | ☐ Crops and/or livestock |
| ☒ | ☐ Ecosystems |
| ☒ | ☐ Any other significant area |

## Experiments of concern

Does the work involve any of these experiments of concern:

| No | Yes |
|-----|-----|
| ☒ | ☐ Demonstrate how to render a vaccine ineffective |
| ☒ | ☐ Confer resistance to therapeutically useful antibiotics or antiviral agents |
| ☒ | ☐ Enhance the virulence of a pathogen or render a nonpathogen virulent |
| ☒ | ☐ Increase transmissibility of a pathogen |
| ☒ | ☐ Alter the host range of a pathogen |
| ☒ | ☐ Enable evasion of diagnostic/detection modalities |
| ☒ | ☐ Enable the weaponization of a biological agent or toxin |
| ☒ | ☐ Any other potentially harmful combination of experiments and agents |

