## [Peer Review File · Nature Genetics]

Peer Review Information

Manuscript Title: Telomere-to-telomere Citrullus Super-pangenome Provides Direction for Watermelon Breeding

Corresponding author name(s): Dr Hang He

Editorial Notes:

Transferred manuscripts This document only contains reviewer comments, rebuttal and decision letters for versions considered at Nature Genetics.

Reviewer Comments & Decisions:

Decision Letter, initial version:

22nd Sep 2023

Dear Dr He,

Your Article, "A T2T Citrullus Super-pangenome Provides Direction for Watermelon Breeding" has now been seen by 3 referees. You will see from their comments copied below that while they find your work of considerable potential interest, they have raised quite substantial concerns that must be addressed. In light of these comments, we cannot accept the manuscript for publication, but would be interested in considering a substantially revised version that fully addresses these serious concerns.

We hope you will find the referees' comments useful as you decide how to proceed. If you wish to submit a substantially revised manuscript, please bear in mind that we will be reluctant to approach the referees again in the absence of major revisions.

To guide the scope of the revisions, the editors discuss the referee reports in detail within the team with a view to identifying key priorities that should be addressed in revision. As you will see from these comments, Reviewer #1 notes that two of the genomes are misidentified as to species (see Reviewer #1 Review Attachment for details), which will largely affect many of the analyses. Reviewers #2 and #3 have also identified important aspects of the analyses and the presentation that need to be improved. In addition, please include a comparison with ref.20 from Wu et al., and clearly clarify the novelty and advance of your study. We hope that you will find the prioritized set of referee points to be useful when revising your study.

If you choose to revise your manuscript taking into account all reviewer and editor comments, please

highlight all changes in the manuscript text file. At this stage we will need you to upload a copy of the manuscript in MS Word .docx or similar editable format.

*2) If you have not done so already please begin to revise your manuscript so that it conforms to our Article format instructions, available here. Refer also to any guidelines provided in this letter.

Please be aware of our guidelines on digital image standards.

[redacted]

If you wish to submit a suitably revised manuscript we would hope to receive it within 6 months. If you cannot send it within this time, please let us know. We will be happy to consider your revision so long as nothing similar has been accepted for publication at Nature Genetics or published elsewhere. Should your manuscript be substantially delayed without notifying us in advance and your article is eventually published, the received date would be that of the revised, not the original, version.

Thank you for the opportunity to review your work.

Sincerely,
Wei

Wei Li, PhD
Senior Editor
Nature Genetics
New York, NY 10004, USA
www.nature.com/ng

Reviewers' Comments:

Reviewer #1:

Remarks to the Author:

Zhang et al. here construct a *Citrullus* super-pangenome by de novo assembling 27 telomere-to-telomere gap-free genomes that include all seven watermelon species and *C. lanatus* var. "cordophanos" [the correct spelling is cordophanus]. Earlier this year, Wu et al. published another super-pangenome for which they re-sequenced 201 wild watermelon accessions and then combined these data with previously published genomes (from Guo et al., 2019) for a total of 547 watermelon accessions including 349 *C. lanatus* (243 cultivars, 88 landraces and 18 *C. lanatus* subsp. cordophanus), 31 *C. mucosospermus*, 131 *C. amarus* and 36 *C. colocynthis*. Zhang et al.'s study differs by including 1 accession of *C. rehmi*, 1 of *C. naudinianus*, and 1 of *C. ecirrhosus*. Below I have pasted a table showing the precise overlap in germplasm sampling btw. the two studies.

Unfortunately, PI 254622 and PI 288522 are misidentified as to species in this new study, which immediately jumped out to me in Fig. 1, a phylogenetic tree of the 27 germplasm. Many of the analyses will therefore need to be redone.

In terms of the conclusions, it's hard to see what the new study contributes that wasn't found by Wu et al. (2023), and it behooves Zhang et al. to explain where their T2T genomes lead to insights not already reported by Wu et al.

As an aside, the name of the sole non-Chinese author on this study, Robert L. Jarett, is here given as Bob Jarett, which makes me suspect that Dr. Jarett did not see the submitted version, contrary to the claim that "B.J. provided plant materials, suggestions for the project, and edited the manuscript."

Reference cited:

Wu, S., Sun, H., Gao, L., Branham, S., McGregor, C., Renner, S. S. Xu, Y., Kousik, C., Wechter, W. P., Levi, A., and Z. Fei. 2023. A *Citrullus* genus super-pangenome reveals extensive variations in wild and cultivated watermelons and sheds light on watermelon evolution and domestication. *Plant Biotechnology Journal*, <https://doi.org/10.1111/pbi.14120>

Zhang et al. here construct a Citrullus super-pangenome by de novo assembling 27 telomere-to-telomere gap-free genomes that include all seven watermelon species and *C. lanatus* var. "cordophanos" [the correct spelling is cordophanus]. Earlier this year, Wu et al. published another super-pangenome for which they re-sequenced 201 wild watermelon accessions and then combined these data with previously published genomes (from Guo et al., 2019) for a total of 547 watermelon accessions including 349 *C. lanatus* (243 cultivars, 88 landraces and 18 *C. lanatus* subsp. cordophanus), 31 *C. mucospermus*, 131 *C. amarus* and 36 *C. colocynthis*. Zhang et al.'s study differs by including 1 accession of *C. rehmi*, 1 of *C. naudinianus*, and 1 of *C. ecirrhosus*. Below I have pasted a table showing the precise overlap in germplasm sampling btw. the two studies.

Unfortunately, PI 254622 and PI 288522 are misidentified as to species in this new study, which immediately jumped out to me in Fig. 1, a phylogenetic tree of the 27 germplasm. Many of the analyses will therefore need to be redone.

In terms of the conclusions, it's hard to see what the new study contributes that wasn't found by Wu et al. (2023), and it behooves Zhang et al. to explain where their T2T genomes lead to insights not already reported by Wu et al.

As an aside, the name of the sole non-Chinese author on this study, Robert L. Jarett, is here given as Bob Jarett, which makes me suspect that Dr. Jarett did not see the submitted version, contrary to the claim that "B.J. provided plant materials, suggestions for the project, and edited the manuscript."

Germplasms analyzed by Zhang et al.	Taxon name	Geography	Analyzed by Wu et al. 2023, along with some 500 other germplasms
PI 596694 (male)	C. naudinianus	Namibia	Species not incl.
PI 525081	C. colocynthis	Egypt	Other PI used
PI 632755	C. colocynthis	Namibia	632755
PI 652554	C. colocynthis	India	652554
PI 537300	C. colocynthis	Turkmenistan	537300
PI 670011	C. rehmi	Namibia	Species not incl.
PI 673135	C. ecirrhosus	Namibia	Species not incl.
PI 482276	C. amarus	Zimbabwe	482276
PI 296341-FR	C. amarus	South Africa	296341
PI 271769	C. amarus	South Africa	271769
PI 189225	C. amarus	Congo	189225
RCAT 055816	C. amarus?really?	Hungary?	Other PI used
PI 532732	C. mucospermus	Congo	Other PI used
PI 595203	C. mucospermus	Nigeria	595203

PI 254622	C. lanatus landrace. This PI # is cordophanus	Sudan	254622
HeiShanRen	C. lanatus landrace	Ukraine	HeiShanRen
PI 381740	C. lanatus landrace	India	381740
DaBanHongZiGua	C. lanatus landrace	China	DaBanHongZiGua
PI 288522	C. lanatus cultivar This PI# is C. amarus	India	288522
SanBaiGua	C. lanatus landrace	China	SanBaiGua
ShiHongNo.2	C. lanatus cultivar	China	ShiHong No.2
Sugarlee	C. lanatus cultivar	United States	Sugarlee
CharlestonGray	C. lanatus cultivar	United States	CharlestonGray
CalhounGray	C. lanatus cultivar	United States	Not included
Allsugar	C. lanatus cultivar	United States	Not included
G42	C. lanatus cultivar	China	Not included
PKR6	C. lanatus cultivar	China	Not included

Wu, S., Sun, H., Gao, L., Branham, S., McGregor, C., Renner, S. S. Xu, Y., Kousik, C., Wechter, W. P., Levi, A., and Z. Fei. 2023. A Citrullus genus super-pangenome reveals extensive variations in wild and cultivated watermelons and sheds light on watermelon evolution and domestication. *Plant Biotechnology Journal*, <https://doi.org/10.1111/pbi.14120>

Reviewer #2:

Remarks to the Author:

The manuscript offers an insightful look into the creation of a first genus level telomere to telomere super pangenome based on 27 gap-free assemblies. The authors' efforts in assembling, annotating, and validating the SVs using current methodologies are commendable. However, there are areas that could benefit from further clarity and elaboration.

1. Synthesis of Findings: While the analysis is robust, it would be beneficial to see a more direct synthesis between the findings and the SVs annotated. For instance, understanding which source possesses certain genes within or outside large inversions/translocations would be insightful.
2. Incorporation of Previous Data: Given the presence of numerous SVs smaller than 50bp, it might be valuable to incorporate data from Guo et al. 2019 and discuss the new variations that have emerged, and which are shared.
3. Clarity in Manuscript: The manuscript is generally clear, but the alternating use of species names and PI numbers can be confusing. It would be helpful if the authors could maintain consistency or provide a clearer reference.
4. Figures: The figures, especially those beyond Figure 1, could benefit from clearer labeling or color coding to help readers understand which accessions belong to which species.
5. Coherence in Sections: The section on "gene gain and loss during watermelon domestications" could be more coherent. It might be beneficial to focus on certain findings and delve deeper into them for clarity.
6. Website and Data Accessibility: The website mentioned seems to be in its preliminary stages. It would be helpful if the authors could ensure that all links are functional, and that the data is easily accessible:
 - a. The browser has only tracks of G42 and is extremely slow if loads at all. It is good that the fasta

and gffs are available for download, but at very slow download rates.

b. On NGDC database, PRJCA017354 contains all raw data for download but it is also very slow to download and access.

The authors should confirm that the assemblies are also deposited in the relevant international repository. All additional codes and scripts should be deposited in a public repository like GitHub.

7. Specific Line Feedback:

- Line 82-83 - Kordofan or cordophanus. There seems to be some inconsistency in their representation.
- Line 87 - clade misspelled.
- Line 94-101 - This section might benefit from a revision for enhanced clarity.
- Line 97 - perennial life history misspelled.
- Line 112 - genomic studies
- Line 152 - The reference to PKR6 is a bit unclear here. It might be helpful to provide more context or refer to the detailed explanation in line 453. Maybe PKR6 should be an outgroup seeing how it contains very large introgressions from other species?
- Line 171 - Extended data 2a - There are only 19 circos plots. Some plots are missing, please verify.
- Line 171 - Extended data 2b - Some plots are missing, please verify. There is a lot of background noise in some of CWR, dot plots, is it due to repetitive sequences? Interestingly *C. naudinianus* has less noise. Maybe use the anchorwave aligner to generate these plots too.
- Line 202 - It would be helpful to provide quantitative data to support the statement (e.g. correlation size vs N LTR)
- Line 204 - The tissues used for various analyses are a bit unclear. Going over methods section, Table S2 and metadata from NGDC, some are listed as tissue for annotation others, just as illumina RNAseq. Number of tissue libraries per accession also isn't uniform. A more detailed breakdown might assist readers.
- Line 258 - The comparison seems to be relative to G42. Considering this is a pan-genome study, perhaps a comparison to Wu et al. 2023 would be more appropriate?.
- Line 263 - In Figure 2b on line 263, the aggregation seems unbalanced. Some species are represented by only one genome, potentially overlooking intraspecific variations. This approach results in a notably high ratio of private genes when compared to previously released genus-level pan-genomes. I recommend using the extended figure 5b as the primary figure, especially since previous studies, like Li 2023, often present data at the accession level. Additionally, for clarity, maintaining color coding for all species in extended figure 5 would be beneficial. I've also noticed potential issues with missing genes from orthogroups, possibly due to diamond cutoffs. Running miniProt on private proteins to query genomic sequences might address these gaps. E.g., *C. rehmi* private versus all comparison.
- Extended figure 5c - Is the large difference between private genes of the male and female *C. naudinianus* interesting to discuss?
- Line 316 - If available, recombination frequency data that includes these large inversions could provide valuable insights to dropping recombination rates across the inversions - leading to linkage drag.
- Line 342 - It seems the data for G42 lacks expression data. Given the wealth of data collected, an expression atlas might be a valuable addition.
- Line 373 - The species on the x-axis of Figure 3a are a bit challenging to follow. Some markers or labels might help.
- Line 377 - It would be beneficial to know which traits are being referred to here and if they could have been detected using data from Guo et al. 2019.
- Line 408-409: For consistency, please use the same designation for species in Figure 5b.
- Line 410 - Do these gene clusters intersect with inversions or translocations when compared to *lanatus*?
- Line 415: The term "most cluster families" is a bit broad. Providing quantitative data might enhance clarity.

- Line 418: You mention there's no significant variation in the number of major TFs. Could you specify the statistical methods employed? Were you examining enrichment? A cursory examination of your data suggests that the count for bHLH is approximately 45 in *amarus*, in contrast to about 145 in other species. Additionally, the count for Ap2 appears to be lower. If these variations aren't statistically significant, consider omitting this section to avoid redundancy.
 - Line 431: It would be insightful to understand how these findings differ from those in Wu et al. 2023.
 - Line 436: Figure 5d, which highlights extremely high correlations: I believe these calculations should be confined to *Lanatus*. It seems there aren't duplications in other species. Making calculations across all species might inadvertently assume that copies don't interact differently across varied genetic backgrounds, an assumption that hasn't been confirmed.
 - Line 445: Regarding the CNV of CITST2, is there any expression data available for both?
 - Line 459-470: This section seems a bit out of context. Consider revising for better flow and clarity.
 - Line 480: In PCA Extended Fig 7, there's a discrepancy between the shapes used in transcriptomic PCA and metabolomic. Please review.
 - Line 506: It might be valuable to discuss the distribution of these metabolites in the other species.
 - Line 533: Some elaboration on the consistency mentioned here would be beneficial. Also, details on the fruit stage at which RT-PCR was carried out might help.
 - Line 568: The expression details of *C. amarus* seem to differ from findings in Guo et al., 2019, suggesting that the mechanisms by which CIBt regulates watermelon fruit bitterness may be more complicated. How do these new results reconcile with the previous statement?
 - Line 604: Given the size of the SVs in Supp table 26, a comparison with data from Guo et al. 2019 might be insightful.
 - Line 604 – referencing Extended table 26, all but two entries are less than 50bp. Is it possible that this analysis could have been conducted using the data from Guo et al. 2019? It would be beneficial to address this comparison in the discussion. The true strength of the super-pan-genome lies in its absence of reference bias. Highlighting an instance, such as a syntenic block moving to a different location, could effectively illustrate this concept.
 - Line 608: In Figure 7, the annotation of "CA" and "CC" versus all PI IDs is a bit challenging to discern. Consider revising for clarity.
 - Line 646: The reference to "cluster 1" seems out of context. Please provide more details.
 - Line 650: It would be helpful to know where the "Three major chromosomal rearrangements" are located.
 - Line 681-683: The QTL here is its first mention. Elaborating on how the QTL was narrowed down might provide more context.
 - Line 1377: In Figure 4g, annotating the SVs for identification in Table S17 would be beneficial.
 - Line 1405: The reference to Grif16376 is a bit unclear as it's the first mention. Some context would be helpful.
- In conclusion, the manuscript offers valuable insights, but there are areas that could benefit from further refinement. I believe that with these adjustments, the paper will be a significant contribution to the field.

Reviewer #3:

Remarks to the Author:

The manuscript by Zhang et al. reports 27 telomere-to-telomere wild and cultivated watermelon genomes followed by the construction of a pan-genome. They further identified extensive variation among these genomes and provides candidates for functionally important genes. These genomic resources provide an amazing dataset for watermelon research community; however, there are some critical issues that need to be addressed.

- The title is good but "T2T" is not a well-recognized abbreviation which needs to be fully spelled when firstly used. The "T2T" in the abstract should also be modified.
- The current abstract is composed of results-like descriptions and does not give the reader a direct impression the conclusion/impact/biological or genomic insights from this study. Examples include: 1) Ln 31 "35,546,180 SNPs and 461,987 SVs were identified." This is purely a presentation of results. 2) Ln 33 "The SVs identified in *C. lanatus* were inherited not only from *C. lanatus* subsp. *cordophanus*, but also from *C. mucosospermus*." I cannot easily get what this sentence could mean to watermelon breeding or biological studies. 3) Ln 34 "Multi-disease resistance was successfully introduced from the wild relatives into *C. lanatus*, as confirmed by genome analysis." From this description one cannot obtain any new information from the genomic analysis presented in this study. In addition, the botanical taxonomy of watermelon and *Citrullus* genus should be indicated since not all researchers are aware of that.
- Overall, the manuscript is generally written clear but there are also some sentences and descriptions that are difficult to understand or could be further improved for clarity/nativeness. For examples: Ln 36 "to be used in efforts to", verbose; Ln 39 "the cultivated crop", it should be "watermelon"; Ln 59 "In the Americas", it should be "In American countries"; Lns 70-93, it will be more concise to merge these two paragraphs for the introduction of the six wild species in the genus *Citrullus*.; Ln 112, "studies" should be "studies"; Ln 131, "in that it", grammar; Ln 136, "entirety", I guess you mean "landscape"; Ln 137, "occurrence", occur in what? Ln 528, two sentences cannot be linked by a comma. Note that these are only a portion of similar issues.
- Ln 143, the authors first claimed that they selected 27 accessions from the phylogeny and then "also selected" several other accessions with disease resistance. Are these additional species included in the 27 accessions? The authors need to clarify this.
- Ln 158, the authors claimed that an average of 799Gb HiFi data were generated. This is impossible for a mean sequence depth of $81 \times (0.38 \text{ Gb} \times 81 = 30.78 \text{ Gb})$.
- Ln 167, the authors assembled Hi-C-based chromosome-level assemblies for seven accessions, while the ext data fig. 1d-I only contains six genomes.
- Ln 200, this sentence lacks statistical details and/or supplementary tables/figures.
- Ln 229, I have no idea what Cr2 is. What are the species names of PKR6, PI 652554 and PI 537300? It would be interesting to compare the centromere monomer-based phylogeny with the whole-genome gene-based phylogeny. Is it possible to compare the centromeric sequences among the 27 genomes? These will generate more insights into the evolution of centromeric sequences in the *Citrullus* genus.
- Ln 245, the URL has a space before the "post".
- Ln 261, there are only three categorizes, namely core, dis and private, in Fig. 2b, lacking soft-core.
- Ln 286, "basic life processes", I guess you mean "fundamental biological processes".
- Ln 296, It is unclear how the authors compared the 28 genomes. My understanding is that the authors align their 27 genomes to the G42 reference genome and identified over 2 million SVs. The authors will need to clarify this.
- Ln 305, "57 large SVs", how large?
- Ln 308-311, the "By mapping" sentence is too long and should be rephrased.
- The y-axis of fig. 3d should be "SV density" but not "SV Desity".
- Ln 314, "preserved", I guess the authors mean "retained".
- Ln 357, why can these two rearrangements lead to reproductive isolation? Is there any previous work that can support this? If no the authors will need to weaken their expression.
- Lns 373-383, the authors investigated the derived or ancestral states of SVs in different watermelon species and presented some examples. However, this section lacks a conclusion. What can these results mean to watermelon domestication?
- In Fig. 5e, it is very difficult to observe the association between CNVs and the phenotype. All peaks seem to be contributed by SNPs.
- Ln 436, is there any figures or statistics to indicated the "extremely strong correlation"?
- Lns 448-470, these two paragraphs seem to introduce the inbred line "PKR6", which is less pertinent to gene gain or loss. Perhaps the authors could make it an additional section if necessary.

- Ln 598, the authors pinpointed several genes based on selective sweep analysis and identified surrounding variants. However, the first two genes do not seem to be relevant to sugar accumulation and flesh color.

Author Rebuttal to Initial comments

PRJReviewers' Comments:

Reviewer #1:

Reviewer #1:

Remarks to the Author:

Zhang et al. here construct a *Citrullus* super-pangenome by de novo assembling 27 telomere-to-telomere gap-free genomes that include all seven watermelon species and *C. lanatus* var. “cordophanos” [the correct spelling is cordophanus]. Earlier this year, Wu et al. published another super-pangenome for which they re-sequenced 201 wild watermelon accessions and then combined these data with previously published genomes (from Guo et al., 2019) for a total of 547 watermelon accessions including 349 *C. lanatus* (243 cultivars, 88 landraces and 18 *C. lanatus* subsp. cordophanus), 31 *C. mucospermus*, 131 *C. amarus* and 36 *C. colocynthis*. Zhang et al.’s study differs by including 1 accession of *C. rehmi*, 1 of *C. naudinianus*, and 1 of *C. ecirrhosus*. Below I have pasted a table showing the precise overlap in germplasm sampling btw. the two studies.

Unfortunately, PI 254622 and PI 288522 are misidentified as to species in this new study, which immediately jumped out to me in Fig. 1, a phylogenetic tree of the 27 germplasm. Many of the analyses will therefore need to be redone.

In terms of the conclusions, it’s hard to see what the new study contributes that wasn’t found by Wu et al. (2023), and it behooves Zhang et al. to explain where their T2T genomes lead to insights not already reported by Wu et al.

As an aside, the name of the sole non-Chinese author on this study, Robert L. Jarett, is here given as Bob Jarett, which makes me suspect that Dr. Jarett did not see the submitted version, contrary to the claim that “B.J. provided plant materials, suggestions for the project, and edited the manuscript.”

Reference cited:

Wu, S., Sun, H., Gao, L., Branham, S., McGregor, C., Renner, S. S. Xu, Y., Kousik, C., Wechter, W. P., Levi, A., and Z. Fei. 2023. A *Citrullus* genus super-pangenome reveals extensive

variations in wild and cultivated watermelons and sheds light on watermelon evolution and domestication. *Plant Biotechnology Journal*, <https://doi.org/10.1111/pbi.14120>

Response:

We would like to express our heartfelt gratitude for your thoughtful review of our manuscript now entitled "Telomere-to-telomere *Citrullus* Super-pangenome Provides Direction for Watermelon Breeding." Your feedback is immensely valuable, and we are committed to addressing your comments and concerns effectively.

The misspelling of "*cordophanus*" has been corrected in the revised manuscript. Thank you for pointing this out.

One of the key contributions of our research is the establishment of telomere-to-telomere gap-free reference genomes for all species within the *Citrullus* genus. This accomplishment marks a significant advancement in watermelon genomics as it provides the *Citrullus* genus with the highest quality reference genomes for all of its species. Our study also generated complete mitochondrial and chloroplast genome assemblies for all species. These assemblies represent a comprehensive resource that facilitates a more precise and in-depth analyses of the genomic architecture, gene content, and specific structural variations in and between *Citrullus* species. These data provide a foundation for future research in this field.

Distinguishing Our Study from Wu et al. (2023):

We acknowledge your inquiry about the distinctiveness of our study in comparison to Wu et al. (2023). While both studies address watermelon genomics, our work contributes uniquely in the following ways:

a. Comprehensive Reference Genomes:

Our study presents telomere-to-telomere gap-free reference genomes for all *Citrullus* species, including those omitted in the study of Wu et al. (2023). The genome assemblies of *C. ecirrhosus*, *C. naudinianus* and *C. rehmii* (not included in Wu et al.) provide significant new information that has not previously, and that is not currently, available. The contig numbers of the three new assemblies of Wu et al. were 77 for *C. mucosospermus* USVL531-MDR, 38,258 for *C. amarus* USVL246-FR2, and 15,928 for *C. colocynthis* PI 537277, while that of other three reported assemblies that were included in their manuscript were 367 for *C. lanatus* 97103, 21,498 for *C. lanatus* Charleston Gray, and 86 for *C. lanatus* subsp. *cordophanus*. Distinctly, in our study, of all 27 assemblies, no gaps among contigs remains. These reference genomes will be useful in future pangenomic and population genetic/genomic studies with these three taxa. Compared to the study of Wu et al., with the number of assembled genomes increased from 6 to 27, the inclusion of all 7 *Citrullus* species and the gap-free nature of our reference genomes are essential features of our study that allow for a more comprehensive and accurate comparative analyses of all species.

b. Super-Pangenome Analysis:

We adopted a truly super-pangenome (genus-wide) approach that is more comprehensive than previous studies. In the study of Wu et al., 13,256,154 SNPs and 2,277,760 small indels were identified among 4 species. However, in our study, with the 27 T2T assembled genome sequences of 7 species, 35,546,180 SNPs and 461,987 structural variations were identified. And compared to the study by Wu et al., our super-pangenome elucidated an additional 8736 novel gene families, with 4913 gene families being contributed by the *C. rehmii*, *C. ecirrhosus*, *C. naudinianus* species. This approach allowed us to explore the collective gene content and genomic diversity within the *Citrullus* genus in a novel and detailed manner. Only one large inter-chromosomal rearrangement involving Chr01 and Chr04 in *C. colocynthis* relative to three other *Citrullus* species (*C. lanatus*, *C. mucospermus* and *C. amarus*) was reported by Wu. Our super-pangenome provided other two major chromosomal rearrangements among species in *Citrullus*, including one involving Chr11 and Chr02 in *C. naudinianus* relative to the six other *Citrullus* species and another involving Chr01 and Chr09 in *C. rehmii* relative to the other six *Citrullus* species. And our data showed the SVs identified in *C. lanatus* were inherited not only from *cordophanus*, but also from *C. mucospermus*, suggesting that *cordophanus* was not the only ancestor of cultivated watermelon. Finally, our results shed light on gene gain and loss dynamics, especially the lost disease-resistant genes were characterized and had huge potentials to be utilized during further breeding, that previous studies did not discuss.

c. Functional Implications:

Our work delves into the functional aspects of the pangenome, including transcriptomic and metabolomic analysis among *Citrullus* species, revealing genes' SV distributions and expression levels that could possibly related to multiple agronomic traits like bitterness, sweetness, flesh color, fruit ripening, fruit size, seed size and disease resistance in wild and cultivated watermelon accessions. The comprehensive study offers insights into potential trait variations and adaptations specific to *Citrullus* species. This aspect was not extensively explored in Wu et al. (2023).

We have revised the manuscript to explicitly highlight these unique aspects of our study and clarify where our contributions differ from prior research.

Clarification of PI 254622 and PI 288522 Classification:

We appreciate your concern regarding the classification of PI 254622 and PI 288522 in our study. We want to emphasize that our classification is rooted in substantial evidence, including extensive genomic data. These accessions are both listed as *Citrullus lanatus* (Thunb.) Matsum. & Nakai in the GRIN database PI 254622 GRIN-Global (ars-grin.gov), PI 288522 GRIN-Global (ars-grin.gov). Numerous studies conducted by researchers at the University of Florida, North Carolina State University, and assessments by the Cucurbit Crop Germplasm Committee have consistently supported these classifications. Both PI 254622 and PI 288522 were noted as *Citrullus lanatus* in the papers authored by Guo et al. (2019) and Wu et al. (2019).

We have listed all the accessions utilized in our study at the species level. Classification of *Citrullus* at the levels of subspecies or variety are not currently used in the GRIN database for the materials we have listed. Our data do support that PI 254622 can be classified as *C. lanatus* var. *cordophanus* or *C. lanatus*

subsp. *cordophanus*, and this change to the classification of PI 254622 in the GRIN database is currently pending.

We acknowledge the updated classification shift for PI 288522 in the publication by Renner et al. (2021), and then later by Wu et al. (2023). However, our data strongly support the classification of PI 288522 as *Citrullus lanatus*. This is supported by the image of seeds in the GRIN database (below) and phenotypic examination during the course of our study.

Figure R1. Seed photo of PI 288522 in GRIN database. (We have the same seed morphology of PI 288522 used in our study.)

While we are not botanists, we are confident that our classification is accurate based on our analysis. We will ensure that the manuscript explicitly outlines the supporting data and methodology where necessary.

Note:

Regarding the questioned origin of RCAT 055816, this accession was indeed collected in Hungary as detailed in the publication by Tóth Zoltán et al. (2007). Its previous classification of *Citrullus lanatus* var. *citroides* has since been updated to *Citrullus amarus*. Our genomic data and field observations of its morphology unequivocally affirm that RCAT 055816 is *Citrullus amarus*. We included this accession due to its remarkable stress tolerance and exceptional plant vigor, making it a valuable candidate for the genetic enhancement of sweet watermelon.

#	Cultivars	Short name	Latin name	Code (Tápiószele)
1	Finn 168	Fin.	Citrullus colocynthis	RCAT036168
2	Belga 172	Bel.	Citrullus colocynthis	RCAT036172
3	Portugál 547	Prt.	Citrullus colocynthis	RCAT035547
4	Szeged 099	Szg.	Citrullus l. citroides	RCAT036099
5	De Bánát 235	Rom.	Citrullus l. citroides	RCAT035235
6	Újszilvás 816	Újs.	Citrullus l. citroides	RCAT055816
7	Bácsbokod 917	Bác.	Citrullus l. lanatus	RCAT035917
8	Napsugár 257	Nap.	Citrullus l. lanatus	00257/05
9	Sándorfalva 105	Snd.	Citrullus l. lanatus	RCAT036105
10	Dévaványa 101	Dév.	Citrullus l. lanatus	5101/02
11	Bentsegyfalva 000	Bta.	Citrullus l. lanatus	0000/05

Figure R2. Copy of Table 1 of Tóth Zoltán et al. (2007) paper.

We appreciate your comments regarding Dr. Robert Jarret. His name was inadvertently presented as Bob Jarret, and we have rectified this error in the revised manuscript. Dr. Jarret played a pivotal role in the research we described and contributed significantly to the manuscript's revisions. Please feel free to contact him (bob.jarret@usda.gov) directly for verification of his involvement and contributions to this work. Please see below the email exchange between Dr. Xingping Zhang and Dr. Robert Jarret.

Revised

Jarret, Bob - REE-ARS <bob.jarret@usda.gov>

8/15/2023 1:21 AM

To: 465

Citullus-0810.v3.docx
223.06 KB

Xingping,

Here are the revisions.

I'm using the new computer today – and I was having a difficult time getting the WORD track changes function to work correctly. So, not all changes are obvious.

I went over the References twice and modified as necessary. Completed a few that were missing info and made some format corrections. Please note the comments on possible missing references or listed references not cited in the text. See also next paragraph.

On the Resources Table – I added dates to the authors – when I could. Some are missing because I wasn't able to identify the correct reference in the References. This might account for references not cited as noted earlier. These author, dates in the Resources Table should probably be replaced with the citation number [] after all the references are complete and numbered.

If you have any questions about any of the changes, let me know.

The manuscript is looking good. One more round should be sufficient to have it ready for submission.

Bob

Figure R3. An example of the e-mail exchange between Dr. Xingping Zhang and Dr. Robert Jarret

Once again, we extend our sincere gratitude for your review and constructive comments. Your insights are instrumental in elevating the quality and relevance of our manuscript.

Reference:

Guo, S., et al. Resequencing of 414 cultivated and wild watermelon accessions identifies selection for fruit quality traits. *Nat. Genet.* 51, 1616–1623 (2019).

Renner, S.S., et al. A chromosome-level genome of a Kordofan melon illuminates the origin of domesticated watermelons. *Proc. Natl. Acad. Sci. USA* 118, e2101486118 (2021).

Wu, S., et al. Genome of 'Charleston Gray', the principal American watermelon cultivar, and genetic characterization of 1365 accessions in the U.S. National Plant Germplasm System watermelon collection. *Plant Biotechnol. J.* 17,2246–2258 (2019).

Wu S., et al. A *Citrullus* genus super-pangenome reveals extensive variations in wild and cultivated watermelons and sheds light on watermelon evolution and domestication. *Plant Biotechnol. J.* (2023).

Zoltán, T., et al. Watermelon (*Citrullus l. lanatus*) production in Hungary from the Middle Ages (13th century). *Hungarian Agricultural Research*, 4, 14-19 (2007).

Reviewer #2:

Remarks to the Author:

The manuscript offers an insightful look into the creation of a first genus level telomere to telomere super pangenome based on 27 gap-free assemblies. The authors' efforts in assembling, annotating, and validating the SVs using current methodologies are commendable. However, there are areas that could benefit from further clarity and elaboration.

Thank you for confirming the significance of this study. Your suggestions and comments are invaluable in enhancing our manuscript. We will make every effort to address your specific feedback.

1. Synthesis of Findings: While the analysis is robust, it would be beneficial to see a more direct synthesis between the findings and the SVs annotated. For instance, understanding which source possesses certain genes within or outside large inversions/translocations would be insightful.

Response:

Thank you for the suggestion. Although we provided some information of SV related genes in Tables S27-29, further analysis is needed to check genes within or outside large inversions or translocations. In response, we delved deeper into the structural variations (SVs) and specifically identified genes positioned within large inversions and translocations.

GO and pathway enrichment analysis revealed that genes residing within the extensive inversions and translocations were predominantly enriched in cellulose synthesis (Table S17), as evidenced by enriched entries such as GO:0016759 (cellulose synthase activity), GO:0030244 (cellulose biosynthetic process), and PWY-1001 (cellulose biosynthesis). By combining functional genes reported in literature and multi-omics analysis of this study, we found five interesting genes (Table S16). *CIG42_04g0123000* is within chromosomal translocation occurred in *C. colocynthis*, *C. ecirrhosus*, and *C. rehmi*, this gene was reported to be associated with seed size in Guo et al. 2019. The other four are sweetness related genes, they are within large inversions occurred in *Citrullus amarus*, *C. colocynthis*, *C. ecirrhosus*, *C. naudinianus*, and *C. rehmi*. *CIG42_03g0079100* (*CINAC68*) is a NAC transcription factor, positively regulated sugar and IAA accumulation (Wang et al., 2021), *CIG42_11g0098200* is among candidate genes identified via mGWAS associated with fruit taste (Yuan et al., 2023), *CIG42_11g0103000* and *CIG42_11g0094800* are genes identified from transcriptomic and metabolomic analysis, *CIG42_11g0103000* was reported highly expressed in *C. lanatus* (Gong et al., 2021), while *CIG42_11g0094800* is a new identified gene from our transcriptome and metabolome data.

We have incorporated the list of functional genes and enrichment results into the main text for enhanced clarity and insight. We appreciate your valuable feedback and hope this revision addresses your concerns.

2. Incorporation of Previous Data: Given the presence of numerous SVs smaller than 50bp, it might be valuable to incorporate data from Guo et al. 2019 and discuss the new variations that have emerged, and which are shared.

Response:

Thank you for pointing this out. When cross-referencing with the data from Guo et al. 2019, none of the SVs listed in Table S27 were identified in the VCF variant file obtained in the paper. However, using the original data from Guo et al. 2019, we found 16 SVs with the G42 reference genome model but identified an additional 29 SVs using the pan-genome as reference (Table S28). This result suggests that the pan-genome approach is more comprehensive and powerful for detecting genomic variation in a population.

More SVs can be found in pan-genome, for example, all SVs longer than 20bp in the Table S28 were identified only in pan-genome. The advantages of pan-genome analysis include the ability to fully describe the diversity and variation of a species' genome, identify individual or population-specific genes or variants, and discover new genes or gene functions. These advantages reflect the potential of pan-genome analysis to improve the accuracy and completeness of genome data and facilitate the discovery of novel genomic features and insights.

3. Clarity in Manuscript: The manuscript is generally clear, but the alternating use of species names and PI numbers can be confusing. It would be helpful if the authors could maintain consistency or provide a clearer reference.

Response:

We genuinely appreciate your constructive feedback regarding the clarity of our manuscript, especially in terms of maintaining consistency in the use of species names and PI numbers. In response to this, we've taken several steps to enhance the clarity and coherence:

We have implemented color-coding for all figures containing PI numbers, correlating them to their respective species, to provide a visual aid that can guide the reader through the associations between PI numbers and species.

In the manuscript text, we have made a concerted effort to always pair PI numbers with their respective species names (e.g., PI 296341-FR (CA)) to ensure that the readers can always relate the PI numbers to a specific species without referring back to previous sections.

We believe that these adjustments will substantially improve the manuscript's readability and minimize potential confusion among the readers. We hope that these changes address your concerns effectively and enhance the manuscript's clarity and consistency.

4. **Figures:** The figures, especially those beyond Figure 1, could benefit from clearer labeling or color coding to help readers understand which accessions belong to which species.

Response:

Thank you for your feedback regarding the labeling and color coding of the figures in our manuscript. We appreciate your suggestion to improve the clarity for readers in understanding which accessions belong to which species.

In response to your comment, we have revised the figures to include clearer labeling and color coding. Each accession now has a corresponding label or color that indicates its respective species. This modification will help readers easily identify and differentiate the accessions belonging to different species throughout the figures.

We believe that these changes significantly enhance the clarity and comprehensibility of the figures and improve the overall presentation of our research findings. Thank you for bringing this to our attention, and we sincerely appreciate your valuable input.

5. **Coherence in Sections:** The section on "gene gain and loss during watermelon domestications" could be more coherent. It might be beneficial to focus on certain findings and delve deeper into them for clarity.

Response:

Thank you for pointing out the coherence issues in the section on "gene gain and loss during watermelon domestication."

In response, we have restructured the figures to better illustrate our findings. In the initial part, we present examples of gene loss in cultivated watermelons, followed by instances of gene gain in the latter part. Alongside this, we've made textual adjustments to enhance the flow and connection between the discussed points.

We believe these changes will provide readers with a clearer and more coherent understanding of the gene dynamics during watermelon domestication. Following your suggestion, we essentially rewrote this

section in the revised manuscript. We appreciate your valuable feedback and hope this revision addresses your concerns.

6. Website and Data Accessibility: The website mentioned seems to be in its preliminary stages. It would be helpful if the authors could ensure that all links are functional, and that the data is easily accessible:

Response:

Thank you for your feedback regarding the accessibility of the website mentioned in our manuscript. We apologize for any inconvenience caused by the preliminary stage of the website. We understand the importance of ensuring that all links are functional and that the data is easily accessible for readers. We are dedicating every effort and professional expertise to enhance the functionality and data accessibility of our website. Rest assured, we remain committed to the continuous improvement of our website and database.

a. The browser has only tracks of G42 and is extremely slow if loads at all. It is good that the fasta and gffs are available for download, but at very slow download rates.

Response:

We have extended the genome browser functionality to 28 genomes. This should provide a more comprehensive view of the data for readers. We have optimized the download rates of the fasta and gff files to improve the overall speed and efficiency.

b. On NGDC database, PRJCA017354 contains all raw data for download but it is also very slow to download and access.

The authors should confirm that the assemblies are also deposited in the relevant international repository. All additional codes and scripts should be deposited in a public repository like GitHub.

Response:

Thank you for your valuable feedback regarding the accessibility of the NGDC database and the deposition of assemblies, codes, and scripts.

1. We have ensured that the assemblies associated with our study are deposited in the National Center for Biotechnology Information (NCBI). The accession numbers (PRJNA1031825) and relevant information are now included in the manuscript. This will allow readers to access the assemblies directly from the international repository, providing a reliable and accessible source of the data.

2. We have deposited all additional codes and scripts used in our study in the GitHub (<https://github.com/yilinZhang-bio/Watermelon-pangenome>). We have included the repository link and instructions on how readers can access and utilize the codes and scripts for further analysis. This ensures transparency and reproducibility of our research.

By implementing these measures, we aim to improve the accessibility and availability of the raw data, assemblies, codes, and scripts associated with our study.

7. Specific Line Feedback:

- Line 82-83 - Kordofan or cordophanus. There seems to be some inconsistency in their representation.

Response:

Thank you for pointing out the inconsistency in the representation of the species name "Kordofan" or "Cordophanus" in lines 82-83 of the manuscript. We apologize for this error and appreciate your keen observation.

To address this issue, we have made the necessary correction in the manuscript. The correct spelling of the sub-species name is "*cordophanus*." We ensure that this spelling is used consistently throughout the manuscript.

- Line 87 – clade misspelled.

Response:

Thank you for catching the misspelling of the word "clade" in line 87 of the manuscript. We appreciate your attention to detail and apologize for the error. We have corrected the spelling of "clade" in the manuscript to ensure accuracy and clarity.

- Line 94-101 – This section might benefit from a revision for enhanced clarity.

Response:

Thank you for pointing out the clarity issues in the section from lines 94-101.

Upon careful consideration and in the interest of enhancing the overall coherence and clarity of the manuscript, we have decided to remove this section entirely from the text.

We believe that the omission of this section will not detract from the core findings and discussions in our study. We appreciate your discerning feedback, which helps us to improve the manuscript.

- Line 97 – perennial life history misspelled.

Response:

Thank you for identifying the misspelling of "perennial life history" in line 97 of the manuscript. We appreciate your careful review and apologize for the error. We have corrected the spelling of "perennial life history" in the manuscript to ensure accuracy and clarity.

- Line 112 – genomic studies

Response:

Thank you for identifying the misspelling of "genomic studies" in line 112 of the manuscript. We have removed the extra spaces.

- Line 152 – The reference to PKR6 is a bit unclear here. It might be helpful to provide more context or refer to the detailed explanation in line 453. Maybe PKR6 should be an outgroup seeing how it contains very large introgressions from other species?

Response:

Thank you for the suggestion. We have revised the description about the material PKR6 as follows: To investigate how the genome structure changes when introducing resistant traits from wild relatives into watermelon cultivars, we specifically chose the inbred line PKR6. This line was developed by intercrossing a watermelon cultivar with three accessions of *C. amarus* (PI 296341-FR, PI 189225, and PI

482270), along with one accession of *C. mucospermus* (PI 595203). In this process, we incorporated a combination of resistance genes, including those conferring resistance to *Fusarium oxysporum* race 1 and 2, anthracnose, powdery mildew, watermelon mosaic virus, and Zucchini yellow mosaic virus.

- Line 171 – Extended data 2a – There are only 19 circos plots. Some plots are missing, please verify.

Response:

Thank you for bringing the issue with Extended Data 2a to our attention. We apologize for any confusion caused by the missing circos plots.

Upon reviewing our records, we have identified an error in the numbering of the circos plots in Extended Data 2a. There should have been a total of 27 circos plots, and we have added the missing plots. Thank you once again for your valuable feedback.

- Line 171 – Extended data 2b – Some plots are missing, please verify. There is a lot of background noise in some of CWR, dot plots, is it due to repetitive sequences? Interestingly *C. naudinianus* has less noise. Maybe use the anchorwave aligner to generate these plots too.

Response:

We appreciate your careful review and feedback on Extended Data 2b. We apologize for the oversight regarding the missing plots and will ensure their inclusion in the revised manuscript.

The background noise in some of the CWR dot plots could indeed be due to repetitive sequences. We agree that the lower level of noise in the *C. naudinianus* plots could be indicative of fewer repetitive sequences or lesser alignment in this species.

Your suggestion to utilize the Anchorwave aligner is very much appreciated. Anchorwave has been shown to be effective at handling repetitive sequences and could potentially reduce the noise in our plots. We implemented this in our revised analysis to ensure the most accurate and interpretable representation of our data (Extended Figure 3).

- Line 202 - It would be helpful to provide quantitative data to support the statement (e.g. correlation size vs N LTR)

Response:

Thank you for your valuable feedback on our manuscript. We want to show our gratitude to the reviewer for this suggestion. We have calculated the Pearson correlation coefficients of the genome size and length of TEs for all 26 genomes (Figure R4). The genome size was highly correlated with the total repeat size with the Pearson correlation coefficient of 0.91. Compared to *C. lanatus*, the proportion of retroelements of the *C. ecirrhosus* genome was increased from 29.9 % to 34.4 %, and the proportion of DNA transposon of the *C. rehmii* genome was increased from 18.7 % to 22.7 % (Figure R5), which were the two main factors for their genome size expansions.

Figure R4. The correlation analyses of genome size and the length of TEs. The Pearson correlation coefficients were calculated using length (bp) of specific sequences of 26 *Citrullus* genomes.

Figure R5. The proportions of TEs of *C. ecirrhosus*, *C. rehmi* and *C. lanatus*. The mean proportions of TE elements of *C. lanatus* samples were used here.

• Line 204 – The tissues used for various analyses are a bit unclear. Going over methods section, Table S2 and metadata from NGDC, some are listed as tissue for annotation others, just as illumina RNAseq. Number of tissue libraries per accession also isn't uniform. A more detailed breakdown might assist readers.

Response:

Thank you for the observation on Line 204. To address the clarity issue concerning the tissues used, we have taken measures to enhance clarity:

1. We've split Table S2 into two separates. The first now details the data used for genome assembly and annotation, while the table showcases data pertinent to the analysis of the transcriptome and metabolome during fruit development.

2. We've also revised the metadata in NGDC to clearly specify the particular analysis each data set contributes to in our study. A portion of the RNA-seq data was utilized for genome annotation, while the remainder was employed for transcriptomic and metabolomic analyses. We have segregated into two distinct folders on NGDC.

3. It's noted that there is variability in the number of tissue libraries for genome annotation across different accessions. The number of transcriptomes and types of tissues are largely consistent across materials, while slight variations exist in developmental stages. We believe these minor discrepancies in developmental timing are adequately accounted for by our extensive transcriptome data, ensuring robust genome annotation despite these variations.

We trust that our revised approach will aid readers in better understanding our methods and analyses.

• Line 258 - The comparison seems to be relative to G42. Considering this is a pan-genome study, perhaps a comparison to Wu et al. 2023 would be more appropriate?

Response:

Thank you for your insightful comment regarding line 258. We appreciate your suggestion to compare our results with the study by Wu et al. 2023, given its relevance to pan-genome studies.

We downloaded novel gene sequences for each species from the super pan-genome construction by Wu et al. 2023, sourced from <http://cucurbitgenomics.org/v2/ftp/pan-genome/watermelon/super-pangenome/>. For instance, we accessed sequences like *C_mucosopermus_novel_pep.fa.gz* and *C_lanatus_novel_pep.fa.gz*. Notably, *C. lanatus* presented 2,288 novel genes, *C. mucosopermus* presented 583, *C. amarus* presented 1,922, and *C. colocynthis* presented 2,521 novel genes. We combined these novel gene sequences with our gene-based pan-genome, resulting in the construction of orthogroups. Upon analysis, compared with the reference genome and Wu et al. 2023, our pan-genome elucidated an additional 8,736 novel gene families, with 4,913 gene families being contributed by the *C. rehmi*, *C. ecirrhosus*, *C. naudinianus* species. Consequently, we have incorporated these findings in our main manuscript.

Thank you for bringing this to our attention, and we believe this revision will strengthen the overall quality and relevance of our manuscript.

• Line 263 – In Figure 2b on line 263, the aggregation seems unbalanced. Some species are represented by only one genome, potentially overlooking intraspecific variations. This approach results in a notably high ratio of private genes when compared to previously released genus-level pan-genomes. I recommend using the extended figure 5b as the primary figure, especially since previous studies, like Li 2023, often present data at the accession level. Additionally, for clarity,

maintaining color coding for all species in extended figure 5 would be beneficial. I've also noticed potential issues with missing genes from orthogroups, possibly due to diamond cutoffs. Running miniProt on private proteins to query genomic sequences might address these gaps. E.g., *C. rehmi* private versus all comparison.

Response:

Thank you for the comprehensive feedback on Line 263 and the associated figures. The representation of some species by only one genome is due to their limited availability in the USDA watermelon germplasm bank. Unfortunately, these were the only samples accessible for our study. Dr. Robert Jarret, Plant Genetic Resources Conservation Unit of USDA, hopes to get more samples of the species *C. naudinianus*, *C. ecirrhosus* and *C. rehmi* when a future germplasm exploration project to Africa is approved by USDA.

To address your concerns:

We've swapped portions of Figure 2 and Extended Figure 5, as you suggested.

For continuity, we have added consistent color coding across all PI numbers and species, in line with the coding used in Figure 1. This greatly enhances the coherence throughout our paper.

Regarding the potential gaps in gene data, we utilized miniProt to map private genes of 7 species against all genomic sequences of other species. We employed filter parameters set at identity $\geq 70\%$, mapped fraction $\geq 70\%$ and mapped to the same synteny. As a result, 918 of the private genes from *C. lanatus* were found in other species' genomes, 128 from *C. mucosospermus*, 1,020 from *C. amarus*, 954 from *C. colocynthis*, 98 from *C. rehmi*, 37 from *C. ecirrhosus*, and 468 from *C. naudinianus*. Based on this, we have made necessary revisions to our figures and accompanying text.

We truly appreciate your insightful recommendations and believe they have enriched our study's presentation and analysis.

- Extended figure 5c – Is the large difference between private genes of the male and female *C. naudinianus* interesting to discuss?

Response:

Thank you for pointing out the difference in private genes between male and female *C. naudinianus* in Extended Figure 5c. In response to your comment, we utilized miniProt to map the private genes of *C. naudinianus* male onto the female genome and vice versa. After applying specific filters (identity $\geq 70\%$, mapped fraction $\geq 70\%$ and mapped to the same synteny), the number of private genes for *C.*

naudinianus male was revised to 36, and for the female, it changed to 35. This discrepancy might have been introduced during orthogroups identification due to the diamond cutoffs. Based on these updated results, we have made appropriate modifications to Extended Figure 5c.

• Line 316 – If available, recombination frequency data that includes these large inversions could provide valuable insights to dropping recombination rates across the inversions – leading to linkage drag.

Response:

Thank you for providing valuable feedback. Studies have shown that large inversions can suppress recombination by reducing crossing over (Drouaud et al., 2006; Wellenreuther and Bernatchez 2018; Huang et al., 2020; Tang et al., 2022). On Chr11 of the PKR6 accession, two large inversions (4.8 Mb and 2.5 Mb) inherited from *C. amarus* (PI 296341-FR or PI 482270) were identified. Traits associated with linkage drag, such as high levels of marker segregation distortion, low fruit set, and diminished pollen viability, have been observed in mapping populations resulting from crosses between *C. lanatus* and *C. amarus* (Hawkins et al., 2001; Ren et al., 2012; Sandlin et al., 2012; McGregor and Waters 2013; Ren et al., 2014; Ren et al., 2015).

When constructing genetic maps, the recombination frequency data for each chromosome is calculated based on the relationship between the genetic distance (cM) of markers and their physical distance. Ren et al. (2012) utilized *C. amarus* accession PI 296341-FR (an accession of this study) and *C. lanatus* cultivar accession 97103 to construct a high-resolution genetic map. They identified approximately 640 recombination events across 11 chromosomes. The number of recombination events per chromosome ranged from 24 to 88, with an average of 58.45. Chromosomes 08 and 11 had the fewest recombination events, with 28 and 24, respectively, which were significantly lower than the other chromosomes. On the contrary, chromosome 05 had the highest number of recombination events, with 88. This pattern aligns with the large inversions map of PKR6, revealing large inversions on chromosomes 08 and 11 (Fig. 3c; Extended data Fig. 5f), which ultimately lead to reduced recombination rates. Ren et al. (2015) used an F2 population derived from another *C. amarus* accession PI 189225 (an accession of this study) and a cultivated watermelon accession K3 to construct a high-density genetic linkage map. They identified a large segment of approximately 21.2 Mb on chromosome 11 that exhibited an opposite orientation between genetic and physical positions. The T2T gap-free genome assemblies of *C. amarus* and PKR6 also revealed the presence of large inversion on chromosome 11. The recombination rates in this population ranged from 2.0 to 4.2 cM/Mb, with the highest recombination rate observed on chromosome 05 at 4.2 cM/Mb. This is consistent with our structural variation analysis, which clearly indicates that chromosome 05 has very few inversions (Fig. 3c; Extended data Fig. 5f).

The research in the two aforementioned papers clearly demonstrates that large inversions reduce recombination frequency. When introducing desirable traits from *C. amarus* into cultivated watermelon

through backcross breeding, the presence of large inversions on chromosome 11 can lead to severe linkage drag. To avoid this problem, it is necessary to select donor lines without inverted fragments that contain target genes. Tang's research of the pan-genome of potatoes provides an excellent example of Y locus that controls carotenoid content in the tuber (yellow flesh color), that gene was located around 1.5-2 kb proximal to the breakpoint of the 5.8-Mb inversion (Tang et al., 2022).

Based on your valuable suggestion and the literature, we have improved this section of the text as follows:

On Chr11 of the PKR6 accession, two large inversions (4.8 Mb and 2.5 Mb) inherited from *C. amarus* are preserved (Table S15). The large inversions have been reported in the constructed genetic maps (Ren et al., 2015) and have been shown to significantly reduce recombination frequency (Ren et al., 2012; Ren et al., 2015). Introgression genes located near these regions from *C. amarus* through backcross breeding may lead to severe linkage drag of unexpected phenotypes.

- Line 342 – It seems the data for G42 lacks expression data. Given the wealth of data collected, an expression atlas might be a valuable addition.

Response:

Thank you for your perceptive observation and suggestion regarding line 342. The creation of an expression atlas, as you suggested, could indeed be a valuable addition. This would not only fill the current gap in our data for G42 but also provide a comprehensive overview of gene expression across all conditions and time points studied. We have generated the expression data for G42 and construct an expression atlas in the database.

- Line 373 – The species on the x-axis of Figure 3a are a bit challenging to follow. Some markers or labels might help.

Response:

Thank you for your feedback regarding Figure 3a. We understand that the current labeling of the species on the x-axis might be challenging to follow.

On the x-axis, we lacked a marker to label species. We've employed color distinctions for the PI numbers to delineate the species, ensuring a consistent color-species correspondence across all figures, which might make the figure easier to interpret. We appreciate your valuable feedback and the opportunity to improve our manuscript.

- Line 377 – It would be beneficial to know which traits are being referred to here and if they could have been detected using data from Guo et al. 2019.

Response:

Thank you for suggestion about referring to the traits of three displayed genes. We have described the functions of these three genes in the main text and cited the corresponding references. In addition, none of the three variants here have been found in the vcf variant files obtained in the paper Guo et al. 2019. However, the variation in *CIG42_01g0030700* and *CIG42_03g0058200* can be detected in the original second-generation sequencing data files as presented in the Guo et al. 2019 article, using G42 as the reference.

- Line 408-409: For consistency, please use the same designation for species in Figure 5b.

Response:

Thank you for your attention to detail and the suggestion regarding the species designation in Figure 5b on lines 408-409. We have revised Figure 5b to ensure that the species are designated consistently with the rest of the manuscript.

- Line 410 – Do these gene clusters intersect with inversions or translocations when compared to *lanatus*?

Response:

Thank you for the query on Line 410. We surveyed these gene clusters as you suggested.

Upon examination, we found that several of these gene clusters indeed intersect with regions showing inversions or translocations relative to *C. lanatus*. The locations of 6 clusters on *C. amarus*, and 3 clusters for *C. naudinianus* were overlapped with inversions, while 1 cluster on *C. amarus*, 3 clusters on *C. colocythis* and 1 cluster on *C. naudinianus* were overlapped with translocations. We believe that these genomic rearrangements might play a role in the evolutionary dynamics of these gene clusters. We incorporated these findings into the revised manuscript to provide readers with a more comprehensive understanding of the genetic landscape.

Thank you for this valuable suggestion, and we appreciate your keen observation which will undeniably enhance the depth of our research.

- Line 415: The term "most cluster families" is a bit broad. Providing quantitative data might enhance clarity.

Response:

We acknowledge that this term could be interpreted as vague or overly broad, and we appreciate your suggestion to provide more specific, quantitative data. In the revised manuscript, we have revised the term "most cluster families" to "13 cluster families" for clearer visual representation.

- Line 418: You mention there's no significant variation in the number of major TFs. Could you specify the statistical methods employed? Were you examining enrichment? A cursory examination of your data suggests that the count for bHLH is approximately 45 in *amarus*, in contrast to about 145 in other species. Additionally, the count for Ap2 appears to be lower. If these variations aren't statistically significant, consider omitting this section to avoid redundancy.

Response:

Thank you for pointing out the discrepancies in the TF counts mentioned in Line 418. We deeply regret the oversight.

Upon re-examination, we realized that there were indeed inaccuracies in the reported counts for bHLH and Ap2. Specifically, the correct count of *C. amarus* for bHLH should be 145, and of other six species for Ap2, it is 20.

To address your concerns, we have re-analyzed the data across all 27 accessions using the Wilcoxon test, and the results still indicate no significant differences in the counts. Given the results and in the interest of clarity and brevity, we have decided to omit this section from the manuscript.

Once again, we sincerely apologize for the oversight and appreciate your diligent review which has allowed us to correct and improve our manuscript.

Figure R6. The number of Ap2 transcription factor across seven watermelon species.

- Line 431: It would be insightful to understand how these findings differ from those in Wu et al. 2023.

Response:

Thank for the suggestion to make a comparison between two studies. In this study, we observed the correlation between fruit flesh SSC (*CITST2*) and flesh color (*LCYB*), suggesting that sugar accumulation and fruit flesh coloration may have been the result of co-selection during watermelon domestication and improvement. This study also confirmed the result Wu et al. 2023 described: *CITST2* tandem duplication became a predominant allele in landraces and was almost fixed in cultivars, and fruit flesh SSC levels were significantly higher in accessions carrying the *CITST2* tandem duplication compared to the ones with only one copy. In addition, we also checked the relative expression of *CITST2* revealed the good alignment with the fruit flesh SSC.

- Line 436: Figure 5d, which highlights extremely high correlations: I believe these calculations should be confined to *Lanatus*. It seems there aren't duplications in other species. Making calculations across all species might inadvertently assume that copies don't interact differently across varied genetic backgrounds, an assumption that hasn't been confirmed.

Response:

Thank you for your comment and suggestion. According to our data, duplicated copy of *TST2* occurred in *C. lanatus* only, we have edited Figure 5 excluding other species.

- Line 445: Regarding the CNV of *CITST2*, is there any expression data available for both?

Response:

Thank you for asking this. Yes, we have the relative expression data of both genes collected from some *C. lanatus* accessions and provided in Table R1. The value of the correlation coefficient between *CIPHT4;2* and brix is higher than that between *CITST2* and brix.

Table R1. Brix and relative expression data of CIPHT4;2 and TST2 for accessions of C. lanatus				
Accessions	Taxonomy	CIPHT4;2	TST2	Brix
PI500301	C. lanatus cultivar	1.00	1.00	3.8
PI288522	C. lanatus cultivar	0.76	0.90	4
97103	C. lanatus cultivar	1.79	2.52	9
G42	C. lanatus cultivar	2.92	2.54	10
Sugarlee	C. lanatus cultivar	2.65	0.99	9
PKR6	C. lanatus cultivar	0.19	1.90	8
SL3H	C. lanatus cultivar	0.99	2.94	7.1
WCZ	C. lanatus cultivar	1.03	1.92	7.5
CIT268	C. lanatus cultivar	0.41	0.17	7.3
CIT112	C. lanatus cultivar	0.64	1.64	6.1
PI249010	C. lanatus cultivar	0.23	0.55	2.8
PI189317	C. lanatus cultivar	0.09	0.42	2.4

CIT165	C. lanatus cultivar	0.11	1.53	3.4
PI179878	C. lanatus cultivar	0.59	0.35	4
CIT306	C. lanatus cultivar	0.68	3.17	6.2
PI381740	C. lanatus landrace	0.35	1.75	8
Dabanhongzigua	C. lanatus landrace	0.21	0.81	4
Heishanren	C. lanatus landrace	0.75	2.39	7
PI254622	C. lanatus landrace	0.94	0.84	6
PI254624	C. lanatus landrace	0.36	1.47	2.8
LDH	C. lanatus landrace	0.19	0.26	4.3
PI525084	C. lanatus landrace	1.87	1.03	6.5
AKKZW	C. lanatus landrace	0.29	1.26	6.8
PI512404	C. lanatus landrace	0.09	1.10	9.2
CIT66	C. lanatus landrace	0.30	0.91	7.6
PI270144	C. lanatus landrace	0.81	0.38	7.6
GR25	C. lanatus landrace	0.47	2.76	5

• Line 459-470: This section seems a bit out of context. Consider revising for better flow and clarity.

Response:

Thank you for the comment. We have rewritten the paragraphs in the “Gene gain and loss during watermelon domestication” part. We hope that the revised presentation of this section facilitates the discussion and analysis of PRK6 for better understanding.

• Line 480: In PCA Extended Fig 7, there's a discrepancy between the shapes used in transcriptomic PCA and metabolomic. Please review.

Response:

Thank you for pointing out the discrepancy between the shapes used in the transcriptomic and metabolomic PCA in Extended Figure 7 on line 480. In both diagrams, we've revised the shapes representing developmental stages for uniformity. We understand that maintaining consistency in the visual elements used in figures is crucial for clarity and ease of interpretation.

- Line 506: It might be valuable to discuss the distribution of these metabolites in the other species.

Response:

Thank you for your valuable suggestion on line 506. We agree that discussing the distribution of these metabolites in other species could provide additional context and enrich our study.

Regarding the distribution of sucrose, the cultivated watermelon species (*C. lanatus*) generally exhibits higher content, particularly in samples G42 and Sugarlee, resonating with their characteristic sweetness. In contrast, wild species, including *C. mucospermus*, *C. amarus*, *C. rehmi*, and *C. colocynthis*, display varied sucrose levels. For instance, the *C. lanatus* landrace sample PI 381740 has notable sucrose content, while the *C. amarus/colocynthis/rehmi* species has less. Across developmental stages A-E, spanning from 10 to 42 days, a peak in sucrose accumulation is evident around days 18 and 26 for specific samples.

For cucurbitacins, *C. colocynthis* species, especially PI 537300 and PI 632755, show heightened levels, aligning with their inherent bitterness. A similar trend of peak accumulation around 18 to 26 days is observed in some samples.

We revised our manuscript to capture these metabolic variations succinctly, enhancing the clarity and depth of our findings. Your feedback is highly appreciated.

- Line 533: Some elaboration on the consistency mentioned here would be beneficial. Also, details on the fruit stage at which RT-PCR was carried out might help.

Response:

Thank you for your suggestion. To check if the expression pattern is consistent with transcriptome data, we selected the fruit stage at which different species showing significant difference for each gene expression. We provided the details in figure legend in the revised manuscript.

- Line 568: The expression details of *C. amarus* seem to differ from findings in Guo et al., 2019,

suggesting that the mechanisms by which *ClBt* regulates watermelon fruit bitterness may be more complicated. How do these new results reconcile with the previous statement?

Response:

Thank you for the reminder and question. By re-analyzing the expression data, we found that there was a mistake in the figure 7 because of the misplacement of accessions of *C. amarus* (PI 482276 and PI 296341-FR) and *C. mucospermus* (PI 595203 and PI 532732), and we have revised that figure. As we can see from Table R2, the expression of *ClBt* was not detected in fruit of both *C. lanatus* and *C. amarus*. Although *ClBt* was reported to be a tissue-specific cucurbitacin regulator, it seems to express higher in leaf than fruit, with near zero FPKM values in fruit samples of most accessions. Collectively, the expression of *ClBt* identified in this study were highly related to previous results. In addition, the expression data of *ClBt* in leaf and fruit at different development stages of more accessions (Table R2), which was not studied by Guo et al., 2019, were also supplied.

Table R2. Expression (FPKM) of *ClBt* in different accessions.

Accession	Taxonomy	Fruit flesh					Leaf	Data resource
		10 DAP	18 DAP	26 DAP	34 DAP	42 DAP		
G42	C. lanatus cultivar	0	0	0	0	0	0	
Sugarlee	C. lanatus cultivar	0	0	0	0	0	0	
DaBanHong ZiGua	C. lanatus landrace	0	0	0	0	0	0	
PI 381740	C. lanatus landrace	0	0	0	0	0	0	This study
PI 595203	C. mucospermus	0	0.3451	1.08378	0.72831	5.63578	2.537	
PI 532732	C. mucospermus	0	0	0	0	0	0	
PI 482276	C. amarus	0	0	0	0	0	5.709	

PI 296341-FR	C. amarus	0	—	0	0	0	8.125	
PI 537300	C. colocynthis	0.28505	0.51671	0	0.73206	0.07193	51.13	
PI 632755	C. colocynthis	3.0543	1.21591	0.81028	—	—	17.97	
PI 670011	C. rehmi	0.10062	0.05181	—	0	—	46.94	
97103	C. lanatus cultivar	0	0	0	0	0	—	
PI 296341-FR	C. amarus	0	0	0	0	0	—	Guo et al., 2019

DAP, days after pollination.

- Line 604: Given the size of the SVs in Supp table 26, a comparison with data from Guo et al. 2019 might be insightful.

Response:

Thank you for your valuable suggestion. Upon comparing the VCF variant file from Guo et al. 2019, none of the SVs listed in Table S28 were identified. However, when utilizing the original second-generation sequencing data from Guo et al. 2019 and employing G42 as a reference, it was revealed that 16 out of the 44 SVs in Table S28 were also present in over 400 watermelon populations (Table S28).

- Line 604 – referencing Extended table 26, all but two entries are less than 50bp. Is it possible that this analysis could have been conducted using the data from Guo et al. 2019? It would be beneficial to address this comparison in the discussion. The true strength of the super-pan-genome lies in its absence of reference bias. Highlighting an instance, such as a syntenic block moving to a different location, could effectively illustrate this concept.

Response:

Thank you for your valuable suggestion. When comparing the VCF variant file from Guo et al. 2019, none of the SVs in Table S28 were found. However, using the original second-generation sequencing data from this paper with G42 as a reference, 16 out of the 44 SVs in Table S28 were identified (Table S28).

For example, the 12 bp DEL in the range of *CIG42_01g0030700*, which is not identified in the VCF file in Guo et al. 2019. However, it was present in both our pan-genome and the variants called with G42 as reference. This indicates that our pan-genome effectively compensates for the previous defects.

Figure R7. Deletion on the gene *CIG42_01g0030700* and its condition in three different watermelon varieties.

• Line 608: In Figure 7, the annotation of “CA” and “CC” versus all PI IDs is a bit challenging to discern. Consider revising for clarity.

Response:

We were sorry about the vague description. We have enriched the figure legend in the revised manuscript to explain the annotation of “CA” and “CC” versus all PI IDs. In addition, we have modified the Figure 7 with specific color of PI IDs indicating different species.

Revision of Figure legend:

Fig. 7 SV Distributions and expression levels of genes that related to agronomic traits in wild and cultivated watermelon accessions.

Overall performance of using SV and differential expression to predict associated genes is shown (see Methods). **Triangles marked with different colors represent the SVs existed in *C. colocynthis* and/or *C. amarus* listed in Table S1.** The gray squares on the left represent the agronomic traits and selective sweeps. Agronomic traits related genes were previously reported, while predicted genes from selective sweeps are newly identified by homologous comparison. The tissues used for expression profiling are indicated at the bottom of each column. **CC, *C. colocynthis*; CA, *C. amarus*.**

• Line 646: The reference to "cluster 1" seems out of context. Please provide more details.

Response:

We were sorry about the vague description. Based on the Neighbor-joining phylogenetic tree of candidate centromere tandem repeat sequences from 26 watermelon accessions, it revealed that monomers aggregate into four clusters. In one branch of the phylogenetic tree (designated as cluster 1), seven watermelon species are included. This suggests that the monomers in cluster 1 are shared and similar among all watermelon species.

- Line 650: It would be helpful to know where the "Three major chromosomal rearrangements" are located.

Response:

Thank you for your valuable suggestion regarding the provision of specific locations for the "Three major chromosomal rearrangements" mentioned on line 650.

In response to your comment, we have added a Table S30 that precisely details the locations of these chromosomal rearrangements. We hope that this addition enhances the comprehensiveness and utility of our manuscript. We appreciate your insightful feedback.

- Line 681-683: The QTL here is it's the first mention. Elaborating on how the QTL was narrowed down might provide more context.

Response:

Thank you for the suggestion. The QTL was named *Qfon1.1* located in a 5 cM region of Chromosome 1 in Ren et al. 2015 paper, we aligned PKR6 genome sequence of the QTL region to G42 and checked the polymorphism of annotated genes. Six polymorphisms were detected between two genomes, the QTL was then narrowed down to a small region with polymorphism in annotated genes (*CIG4201g0002300*, *CIG4201g0002600*, *CIG4201g0004400*). We elaborated this in the revised manuscript.

- Line 1377: In Figure 4g, annotating the SVs for identification in Table S17 would be beneficial.

Response:

Thank you for your valuable suggestion. We agree that annotating the SVs in Figure 4g would greatly enhance the understanding of the data presented in Table S20 (Table S17 in raw version). In the revised

manuscript, we have added annotations to Figure 4g to directly link the SVs to their corresponding entries in Table S20.

• Line 1405: The reference to Grif16376 is a bit unclear as it's the first mention. Some context would be helpful.

Response:

Thank you for pointing out the lack of clarity in the first mention of "Grif16376" on line 1405. Grif16376 is another name we used for *C. rehmii* PI 670011.

To address this issue and enhance the clarity of the manuscript, we have replaced all instances of "Grif16376" with "PI 670011" throughout the manuscript. Specifically, the *C. rehmii* PI 670011 is one of the 27 samples selected for the pan-genome study.

We hope that this revision adequately addresses the concern and improves the overall readability of the manuscript. We appreciate your attention to detail and valuable feedback.

In conclusion, the manuscript offers valuable insights, but there are areas that could benefit from further refinement. I believe that with these adjustments, the paper will be a significant contribution to the field.

Response:

We express our deepest gratitude for your meticulous review and valuable feedback on our manuscript and figures.

Your keen observation and rigorous scrutiny have allowed us to identify and rectify several errors that had previously eluded us. We greatly appreciate your dedication and the thoroughness of your review, which has undoubtedly enhanced the quality and accuracy of our work. We have carefully considered and addressed all the points you raised and believe that the revisions have significantly improved our manuscript.

Thank you once again for your time, expertise, and constructive feedback of our paper and ensuring it makes a significant contribution to the field.

Reference:

Drouaud, J., et al. Variation in crossing-over rates across chromosome 4 of *Arabidopsis thaliana* reveals the presence of meiotic recombination "hot spots". *Genome Res.* 16, 106–114 (2006).

- Gong C., et al. An integrated transcriptome and metabolome approach reveals the accumulation of taste-related metabolites and gene regulatory networks during watermelon fruit development. *Planta*. 254, 35 (2021).
- Guo, S., et al. Resequencing of 414 cultivated and wild watermelon accessions identifies selection for fruit quality traits. *Nat. Genet.* 51, 1616–1623 (2019).
- Hawkins, L.K., et al. Linkage mapping in a watermelon population segregating for fusarium wilt resistance. *J. Amer. Soc. Hort. Sci.* 126:344-350(2001).
- Huang, K and Rieseberg, L. H. Frequency, Origins, and Evolutionary Role of Chromosomal Inversions in Plants. *Front. Plant Sci.* 11:296. doi: 10.3389/fpls.2020.00296 (2020).
- McGregor, C. E. and Waters, V. Pollen viability of F1 hybrids between watermelon cultivars and disease-resistant, infraspecific crop wild relatives. *Hortscience*. 48(12):1428-1432(2013).
- Ren, R., et al. Construction of a high-density DArTseq SNP-based genetic map and identification of genomic regions with segregation distortion in a genetic population derived from a cross between feral and cultivated-type watermelon. *Mol Genet Genomics*. 290:1457-1470(2015).
- Ren, Y., et al. A high resolution genetic map anchoring scaffolds of the sequenced watermelon genome. *PLoS One* 7: e29453 (2012).
- Ren Y., et al. An integrated genetic map based on four mapping populations and quantitative trait loci associated with economically important traits in watermelon (*Citrullus lanatus*). *BMC Plant Biology*. 14:33(2014).
- Ren, Y., et al. Genetic analysis and chromosome mapping of resistance to *Fusarium oxysporum* f. sp. *niveum* (FON) race 1 and race 2 in watermelon (*Citrullus lanatus* L.). *Mol Breed*. 35, 183 (2015).
- Sandlin, K.C., et al. Comparative mapping in watermelon [*Citrullus lanatus* (Thunb.) Matsum. et Nakai]. *Theor. Appl. Genet.* 125:1603-1618(2012).
- Tang, D., et al. Genome evolution and diversity of wild and cultivated potatoes. *Nature*. 606, 535-541(2022).
- Wang, J., et al. The NAC transcription factor CINAC68 positively regulates sugar content and seed development in watermelon by repressing CIINV and CIHG3.6. *Hortic. Res.* 8, 214 (2021).
- Wellenreuther, M. & Bernatchez, L. Eco-evolutionary genomics of chromosomal inversions. *Trends Ecol. Evol.* 33, 427-440 (2018).
- Wu S., et al. A *Citrullus* genus super-pangenome reveals extensive variations in wild and cultivated watermelons and sheds light on watermelon evolution and domestication. *Plant Biotechnol. J.* (2023)
- Yuan, P., et al. Watermelon domestication was shaped by stepwise selection and regulation of the metabolome. *Sci. China Life Sci.* 66, 579–594 (2023).

Zhou, Y., et al. Convergence and divergence of bitterness biosynthesis and regulation in Cucurbitaceae. Nat. Plants 2, 16183 (2016).

Reviewer #3:

Remarks to the Author:

The manuscript by Zhang et al. reports 27 telomere-to-telomere wild and cultivated watermelon genomes followed by the construction of a pan-genome. They further identified extensive variation among these genomes and provides candidates for functionally important genes. These genomic resources provide an amazing dataset for watermelon research community; however, there are some critical issues that need to be addressed.

Response:

Thank you for your summary of our manuscript and for recognizing the value of our work for the watermelon research community. We understand that there are critical issues that need to be addressed, and we are committed to improving our manuscript based on your valuable feedback.

- The title is good but “T2T” is not a well-recognized abbreviation which needs to be fully spelled when firstly used. The “T2T” in the abstract should also be modified.

Response:

Thank you for your valuable suggestion. We have revised the title and the abstract to spell out "Telomere-to-Telomere" in the first instance of usage, and then use the abbreviation "T2T" subsequently. We appreciate your attention to detail and your help in improving the clarity of our manuscript.

- The current abstract is composed of results-like descriptions and does not give the reader a direct impression the conclusion/impact/biological or genomic insights from this study. Examples include: 1) Ln 31 “35,546,180 SNPs and 461,987 SVs were identified.” This is purely a presentation of results. 2) Ln 33 “The SVs identified in *C. lanatus* were inherited not only from *C. lanatus* subsp. *cordophanus*, but also from *C. mucosospermus*.” I cannot easily get what this sentence could mean to watermelon breeding or biological studies. 3) Ln 34 “Multi-disease resistance was successfully introduced from the wild relatives into *C. lanatus*, as confirmed by genome analysis.” From this description one cannot obtain any new information from the genomic analysis presented in this study. In addition, the botanical taxonomy of watermelon and *Citrullus* genus should be indicated since not all researchers are aware of that.

Response:

Thank you for your valuable suggestions and comments. We have revised abstract in the revise manuscript as below. We hope this revision provides clarity of the insights of this study.

To comprehensively decipher the genetic diversity and intricacies within the cucurbit genus *Citrullus*, we generated telomere-to-telomere (T2T) assemblies of 27 distinct genotypes, encompassing all seven *Citrullus* species. The integration of the T2T super-pangenome has expanded the previously published reference genome, T2T-G42, by 399.2 Mb and 11,225 genes. Comparative analysis has unveiled gene variants and extensive structural rearrangements, shedding light on watermelon evolution and domestication processes that enhanced attributes such as bitterness and sugar content while compromising disease resistance. Leveraging intercrossing strategies with wild *C. amarus* and *C. mucospermus*, multi-disease-resistant loci were successfully introduced into cultivated *C. lanatus*, showcasing robust resistance capabilities. The structural variations (SVs) identified in *C. lanatus* have not only been inherited from *cordophanus* but also from *C. mucospermus*, hinting at the presence of additional ancestors beyond *cordophanus* in the lineage of cultivated watermelon. In summary, our investigation significantly enriches the comprehension of watermelon genome diversity and complexity, furnishing comprehensive reference genomes for all *Citrullus* species. This advancement aids in the comprehensive exploration and enhancement of watermelon and its wild relatives for the purposes of crop improvement.

- Overall, the manuscript is generally written clear but there are also some sentences and descriptions that are difficult to understand or could be further improved for clarity/nativeness. For examples: Ln 36 “to be used in efforts to”, verbose; Ln 39 “the cultivated crop”, it should be “watermelon”; Ln 59 “In the Americas”, it should be “In American countries”; Lns 70-93, it will be more concise to merge these two paragraphs for the introduction of the six wild species in the genus *Citrullus*.; Ln 112, “stud ies” should be “studies”; Ln 131, “in that it”, grammar; Ln 136, “entirety”, I guess you mean “landscape”; Ln 137, “occurrence” , occur in what? Ln 528, two sentences cannot be linked by a comma. Note that these are only a portion of similar issues.

Response:

Thank you so much for your thorough review and for pointing out the areas where our manuscript could benefit from improved clarity and nativeness in language use.

We sincerely appreciate the specific examples you provided, which give us clear direction for our revisions. Each point is carefully considered and the manuscript has been revised accordingly:

Ln 36: We have revised the abstract to enhance its completeness.

Ln 39: We have revised the abstract to enhance its completeness.

Ln 59: We have modified "In the Americas" to "In American countries".

Lns 70-93: We merged two paragraphs and refined the main text for clarity.

Ln 112: We have removed the extra space.

Ln 131: We have changed "in that it" to "as".

Ln 136: We have replaced "entirety" with "landscape".

Ln 137: We have replaced "occurrence" with "distribution".

Ln 528: We have changed "but not" to "while they were not observed" to link.

We recognize the importance of clear and concise language. Therefore, aside from the points you mentioned, we have taken a thorough review of the entire manuscript to ensure that the text is clear and well-articulated throughout.

Your keen observations and constructive feedback are invaluable in helping us improve the quality of our work, and we are grateful for the attention to detail you have afforded our manuscript.

- Ln 143, the authors first claimed that they selected 27 accessions from the phylogeny and then “also selected” several other accessions with disease resistance. Are these additional species included in the 27 accessions? The authors need to clarify this.

Response:

Thank you for the comment. Accessions with disease resistance are within in the 27 accessions, “also selected” are misleading. We corrected this in the revised manuscript.

- Ln 158, the authors claimed that an average of 799Gb HiFi data were generated. This is impossible for a mean sequence depth of 81 x ($0.38 \text{ Gb} * 81 = 30.78 \text{ Gb}$).

Response:

Thank you for your careful review and for pointing out this discrepancy on line 158 regarding the average HiFi data generated. To clarify, the 799 Gb HiFi data we referred to is the total amount of data generated, not the average per sample. The average depth of 81x is correct and pertains to each individual sample sequenced. We appreciate your attention to detail and your help in improving our manuscript.

- Ln 167, the authors assembled Hi-C-based chromosome-level assemblies for seven accessions, while the ext data fig. 1d-I only contains six genomes.

Response:

Thank you for pointing out the discrepancy between the text in line 167 and the data presented in Extended Data Figure 1d-I. We have sequenced the Hi-C data for seven accessions, inclusive of PI 537300 and PI 632755, both of which belong to the *C. colocynthis* species. Our supplementary figures primarily spotlight the species genomes; hence, we only utilized the Hi-C heatmap of PI 537300 (*C. colocynthis*).

We value your meticulous attention and assistance in enhancing the accuracy of our manuscript.

- Ln 200, this sentence lacks statistical details and/or supplementary tables/figures.

Response:

Thank you for your valuable feedback on our manuscript. We would like to express our gratitude to you for this valuable suggestion. We have calculated the Pearson correlation coefficients of the genome size and length of TEs for all 26 genomes (Figure R8). The genome size was highly correlated with the total repeat size with the Pearson correlation coefficient of 0.91. Compared to *C. lanatus*, the proportion of retroelements of the *C. ecirrhosus* genome was increased from 29.9 % to 34.4 %, and the proportion of DNA transposon of the *C. rehmii* genome was increased from 18.7 % to 22.7 % (Figure R9), which were the two main factors for their genome size expansions.

Figure R8 The correlation analyses of genome size and the length of TEs. The Pearson correlation coefficients were calculated using length (bp) of specific sequences of 26 *Citrullus* genomes.

Figure R9 The proportions of TEs of *C. ecirrhosu*, *C. rehmii* and *C. lanatus*. The mean proportions of TE elements of *C. lanatus* samples were used here.

- Ln 229, I have no idea what Cr2 is. What are the species names of PKR6, PI 652554 and PI 537300? It would be interesting to compare the centromere monomer-based phylogeny with the whole-genome gene-based phylogeny. Is it possible to compare the centromeric sequences among the 27 genomes? These will generate more insights into the evolution of centromeric sequences in the *Citrullus* genus.

Response:

Thank you very much for asking Cr2 and the species name of PKR6, PI 652554 and PI 537300. Cr2 (TTTAGACACTTTTTAGCCATTCTTTGGTTGGTTTTGAGTTTAGAGGTCATAGTTGTGCATTTTGGAGTTGTTTTTGTAAGTAAAGTGTC) is a centromere tandem repeat identified by telomere-to-telomere gap-free reference genome G42 (Deng et al. 2022). In this study, we identified the same centromere tandem repeat which was found to be the most abundant in all the accessions except PKR6, PI

652554 and PI 537300. PKR6 is *C. lanatus*, PI 652554 and PI 537300 are both *C. colocynthis*. In the revised manuscript we have given the species names of the materials used in this study. Based on your constructive suggestion, we have compared centromeric sequences on each chromosome among the 27 genomes (Extended Data Fig. 5b).

Extended Data Fig. 5b. The comparison of centromeric sequences on each chromosome among the 27 genomes.

Through a comparison between the centromere monomer-based phylogeny and the whole-genome gene-based phylogeny, it becomes evident that, despite *C. naudinianus* being distantly related to *C. lanatus* at the whole-genome level, its centromere sequences, with the exception of chromosome 5, exhibit a closer resemblance to those of *C. lanatus*. Similarly, in the case of *C. amarus* and *C. colocynthis*, the whole-genome analysis suggests a stronger affinity between *C. amarus* and *C. lanatus* than *C. colocynthis*. This trend aligns with the evolutionary relationships identified in the centromere sequences of chromosomes 2, 3, 4, and 6, which demonstrate a significant level of similarity. Conversely, regarding the evolutionary relationships of centromere sequences on the remaining 7 chromosomes, *C. colocynthis* demonstrates a closer relationship to *C. lanatus*. Consequently, it can be inferred that the centromere regions of these 7 chromosomes have evolved independently within the *C. amarus* and *C. colocynthis* species. Based on this difference, we counted the major TE composition of the centromere region of Chr02, Chr03, Chr04, and Chr06 as one class and other chromosomes as another class (Table S12). It is found that these two classes have different TE compositions, and it is these differences that lead to changes in their evolutionary relationships.

Once again, we thank your valuable suggestion to improve our study and manuscript.

- Ln 245, the URL has a space before the “post”.

Response:

Thank you for catching that error on line 245. We have corrected this in the revised manuscript to ensure the URL is accurate and accessible.

- Ln 261, there are only three categorizes, namely core, dis and private, in Fig. 2b, lacking soft-core.

Response:

Thank you for your suggestion regarding line 281. In compliance with Reviewer#2's recommendation, we have relocated Extended Figure 5b to the main figures, while the pan-genome composition of the species has been transferred to the extended figures. In our revised manuscript, we have ensured that the categorization in Figure 2b is consistent with the corresponding text.

- Ln 286, “basic life processes”, I guess you mean “fundamental biological processes”.

Response:

Thank you for your suggestion on line 286. The term "fundamental biological processes" is more precise and appropriate in this context. We have corrected to ensure clarity and scientific accuracy.

- Ln 296, It is unclear how the authors compared the 28 genomes. My understanding is that the authors align their 27 genomes to the G42 reference genome and identified over 2 million SVs. The authors will need to clarify this.

Response:

Thank you for your comment regarding line 296. We apologized if the presentation of our methodology was not clear. You are correct in your understanding. We aligned the 27 genomes to the G42 reference genome and identified over 2 million SVs. To provide clarity, we propose the following revision for line 293-294:

"We compared the 28 genomes by aligning the sequences of the 27 new genomes to the G42 reference genome. "

- Ln 305, "57 large SVs", how large?

Response:

Thank you for your question. The structural variants we validated with an average length of 26,418 bp, confirmed through employment of pair-end PCR amplification.

- Ln 308-311, the "By mapping" sentence is too long and should be rephrased.

Response:

Thank you for your feedback regarding the sentence on lines 308-311. We understand that the sentence may be too long and complex, which could potentially hinder the reader's understanding. Here is our revised sentence:

By mapping all assembly variations and their collinearity with the reference genome, we established a comprehensive SV landscape. This landscape revealed a narrow genetic diversity within cultivated watermelons, but notably greater genetic variation in the wild watermelons (Fig. 3c).

- The y-axis of fig. 3d should be "SV density" but not "SV Desity".

Response:

Thank you for pointing out the error in the y-axis label of Fig. 3d. We have corrected the spelling of "SV density" in the manuscript to ensure accuracy and clarity.

- Ln 314, "preserved", I guess the authors mean "retained".

Response:

Thank you for your suggestion on line 314. We agreed that "retained" might be a more accurate term to use in this context than "preserved". The term "retained" better conveys the idea that the feature or characteristic in question has remained continued to exist.

We have revised the manuscript to replace "preserved" with "retained" on line 314. We appreciate your attention to detail and your effort to improve the clarity and accuracy of our manuscript.

- Ln 357, why can these two rearrangements lead to reproductive isolation? Is there any previous work that can support this? If no the authors will need to weaken their expression.

Response:

Thank you for the constructive suggestions. Chromosomal rearrangements can play a significant role in driving reproductive isolation, impacting hybrid fertility, and reducing interspecific recombination and gene flow (Lowry & Willis 2010; Hou et al., 2014; Baack et al., 2015). Previous studies have reported observations of low fruit set, diminished pollen viability, and decreased recombination frequency in mapping populations resulting from crosses between *C. lanatus* and *C. amarus* (Hawkins et al., 2001; Ren et al., 2012; Sandlin et al., 2012; McGregor and Waters 2013; Ren et al., 2014; Ren et al., 2015). In F1 hybrids between *C. lanatus* and *C. mucospermus*, there was no reduction in pollen viability compared to parental lines. However, hybrids with *C. amarus* exhibited variable pollen viability, ranging from 61.8% to 91.7% (McGregor and Waters 2013), and hybrids with *C. colocynthis* showed pollen viability ranging from 18.5% to 46.0% (Sain and Joshi 2003).

The results of meiotic cytogenetic analysis in interspecific F1 hybrids of *C. lanatus* and *C. colocynthis* indicated the formation of bivalents, ranging from 8.71 to 9.72 II, on average, out of 11 pairs of chromosomes, about 9 pairs were able to associate normally. Such a high bivalent formation in hybrids suggests a close relationship between the two species. Additionally, consistent formation of quadrivalent associations ranging from 0.2 to 0.4 per pollen mother cells indicates that reciprocal translocation inversions may have also contributed to the rearrangement of gene sequences on the chromosomes (Sain et al., 2002). Although the study of Sain et al have indicated the presence of chromosomal rearrangements between the species *C. colocynthis* and *C. lanatus*, the specific chromosomal locations and sizes of these rearrangements were not known. Our research provides precise information on the locations and sizes of these chromosomal rearrangements.

We have integrated your feedback and improved the wording in this paragraph as follows:

As noted in Wu et al.'s findings, we identified a significant inter-chromosomal rearrangement involving Chr01 and Chr04 in *C. colocynthis* compared to three other *Citrullus* species (*C. lanatus*, *C. mucospermus*, and *C. amarus*). This chromosomal rearrangement was confirmed through meiotic cytogenetic analysis conducted on interspecific F1 hybrids of *C. lanatus* and *C. colocynthis*. These alterations in the chromosome structure might contribute to reproductive isolation, impacting hybrid

fertility and reducing interspecific recombination (Hawkins et al., 2001; Sain and Joshi, 2003; Ren et al., 2012; Sandlin et al., 2012; McGregor and Waters, 2013; Ren et al., 2014; Ren et al., 2015), ultimately leading to the differentiation of *Citrullus* species.

- Lns 373-383, the authors investigated the derived or ancestral states of SVs in different watermelon species and presented some examples. However, this section lacks a conclusion. What can these results mean to watermelon domestication?

Response: Thank you very much for pointing this out. In the revised manuscript, we have added this conclusion. The SVs identified in *C. lanatus* were inherited not only from *C. lanatus* subsp. *cordophanus*, but also from *C. mucosospermus*, suggesting at the presence of additional ancestors beyond *cordophanus* in the lineage of cultivated watermelon.

- In Fig. 5e, it is very difficult to observe the association between CNVs and the phenotype. All peaks seem to be contributed by SNPs.

Response:

Thank you very much for this comment. In the case of the *TST2* and *LCYB* genes, previous studies have demonstrated that their associated significant signals of SNPs. In our study, besides the SNP-GWAS, SVs and PAVs were also identified and SV-GWAS and PAV-GWAS were also applied to identify structural variant associated loci. We found that the genomic regions harboring the *TST2* and *LCYB* genes were identified by PAV-GWAS (Figure R10). However, *TST2* gene cannot be identified directly by CNV or PAV-GWAS with second-generation sequencing data, because we found the sequences of two copies were identical.

By comparing SV-GWAS and PAV-GWAS with SNP-GWAS, we found that SNP-GWAS usually had the better statistical power, due to the high density and better identification among population. However, PAV-GWAS still presented some variants that were not detected by SNP-GWAS, although lacking of more linkage support. For example, for the *TST2* gene, SNP-GWAS has significant signal only in Flesh color phenotype, but in PAV-GWAS, this associated 2,368 bp insertion have significant signal in Flesh color, Fruit shape, Rind color, Sugar content on average (SSC.ave) and Sugar content Yanqing (SSC.YQ) (P-value: 1.31e-10, 7.24e-11, 1.00e-07, 4.29e-11, 7.42e-11). Thus, a combination strategy of SV-GWAS, PAV-GWAS and SNP-GWAS would benefit identifying more structural variant associated loci.

Figure R10. GWAS Manhattan plot around *TST2* and *LCYB* gene

Both SNP-GWAS and PAV-GWAS identified *TST2* and *LCYB* gene. (A) GWAS local manhattan plot around *TST2* and *LCYB* gene. (B) Corresponding box plots in accessions carrying distinct alleles. P-values were computed from two-tailed Student's t test. REF, accessions with homozygous reference type of allele; ALT, accessions possessing homozygous alternative allele.

- Ln 436, is there any figures or statistics to indicated the “extremely strong correlation”?

Response:

Thank you for your comment. We apologize for the lack of clarity in our description of the “extremely strong correlation” on line 436. In the revised manuscript, we provided a p-value to quantitatively describe this correlation. We conducted chi-square test and Fisher's exact test on two random variables, *TST2* copy number and *LCYB* genotype. Both sets of results indicate significant difference, which means extremely strong correlation exists between *TST2* copy number and *LCYB* genotype.

- Lns 448-470, these two paragraphs seem to introduce the inbred line “PKR6”, which is less pertinent to gene gain or loss. Perhaps the authors could make it an additional section if necessary.

Response:

Thank you for the comment. We have rewritten the paragraphs in the “Gene gain and loss during watermelon domestication” part. We hope that the revised presentation of this section facilitates the discussion and analysis of PRK6 for better understanding.

- Ln 598, the authors pinpointed several genes based on selective sweep analysis and identified surrounding variants. However, the first two genes do not seem to be relevant to sugar accumulation and flesh color.

Response:

We are sorry about the negligence. This sentence has been modified in the revised manuscript:

“To explore novel candidate genes harboring SVs that may contribute to trait diversity, selective sweeps were made to compare wild and cultivated watermelons.”

Reference:

Baack, E., et al. The origins of reproductive isolation in plants. *New Phytologist*. 207: 968–984 (2015).

Deng, Y., et al. A telomere-to-telomere gap-free reference genome of watermelon and its mutation library provide important resources for gene discovery and breeding. *Mol. Plant* 15, 1268–1284 (2022).

Hawkins, L.K., et al. Linkage mapping in a watermelon population segregating for fusarium wilt resistance. *J. Amer. Soc. Hort. Sci.* 126:344-350(2001).

- Hou, J., et al. Chromosomal rearrangements as a major mechanism in the onset of reproductive isolation in *Saccharomyces cerevisiae*. *Current Biology* 24, 1153–1159(2014).
- Lowry DB, Willis JH. A widespread chromosomal inversion polymorphism contributes to a major life-history transition, local adaptation, and reproductive isolation. *PLoS Biology* 8: e1000500(2010).
- McGregor, C. E. and Waters, V. Pollen viability of F1 hybrids between watermelon cultivars and disease-resistant, infraspecific crop wild relatives. *Hortscience*. 48(12):1428-1432(2013).
- Ren, R., et al. Construction of a high-density DArTseq SNP-based genetic map and identification of genomic regions with segregation distortion in a genetic population derived from a cross between feral and cultivated-type watermelon. *Mol Genet Genomics*. 290:1457-1470(2015).
- Ren, Y., et al. A high resolution genetic map anchoring scaffolds of the sequenced watermelon genome. *PLoS One* 7: e29453 (2012).
- Ren Y., et al. An integrated genetic map based on four mapping populations and quantitative trait loci associated with economically important traits in watermelon (*Citrullus lanatus*). *BMC Plant Biology*. 14:33(2014).
- Sain, R.S. and P. Joshi. Pollen fertility of interspecific F1 hybrids in genus *Citrullus* (Cucurbitaceae). *Curr. Sci.* 85:431–434(2003).
- Sain, R.S., et al. Cytogenetic analysis of interspecific hybrids in genus *Citrullus* (Cucurbitaceae). *Euphytica*.128:205–210(2002).
- Sandlin, K.C., et al. Comparative mapping in watermelon [*Citrullus lanatus* (Thunb.) Matsum. et Nakai]. *Theor. Appl. Genet.* 125:1603-1618(2012).

Decision Letter, first revision:

Our ref: NG-A63345R

8th Jan 2024

Dear Dr. He,

Thank you for submitting your revised manuscript "Telomere-to-telomere *Citrullus* Super-pangenome Provides Direction for Watermelon Breeding" (NG-A63345R). It has now been seen by the original referees and their comments are below. The reviewers find that the paper has improved in revision, and therefore we'll be happy in principle to publish it in *Nature Genetics*, pending minor revisions to satisfy the referees' final requests and to comply with our editorial and formatting guidelines.

Sincerely,
Wei

Wei Li, PhD
Senior Editor
Nature Genetics
New York, NY 10004, USA
www.nature.com/ng

Reviewer #2 (Remarks to the Author):

The authors have addressed all of the concerns that I raised in my initial review. The revisions have been executed with care and have substantially improved the manuscript.

The authors have made a significant effort to clarify the previous ambiguities, strengthen the methodology, and provide a more robust analysis of the data. This has greatly enhanced the overall quality and scientific rigor of the paper.

In terms of novelty and relevance, to the best of my knowledge the findings presented in the manuscript remain novel and contribute meaningfully to watermelon genomic resources.

I believe that the manuscript is now well-suited for publication in Nature Genetics.

A few minor comments:

line 329 the term correlated implies a quantitative analysis, if so mention the value.

line 330 how do these insights help understand the SVs? Their origin? maybe rephrase.

line 451 again the use of strong correlation is vague - write a value.

line 495 you mention correlation but the value in parenthesis is not the correlation coefficient but the r-square

line 556 missing "be" in "seems to expressed" - seems to be expressed

line 558 please write where upstream these deletions are located (e.g kb from translation start site)

Reviewer #3 (Remarks to the Author):

Many thanks for the opportunity to see the revision. The authors have appropriately addressed my concerns. While the manuscript has been improved, there are still some places that could benefit from the improvement of the use of English.

- Ln 4, is Bob Jarret really not Robert Jarret?
- Ln 26, perhaps changing to "by adding 399.2 Mb of sequences and 11,225 genes,"
- Ln 33, suggesting at -> suggesting
- Ln 138-139, change to: The development of PKR6 has introgressed several genes conferring resistance against xxxx
- Ln 220-224, if I understand correctly, change to: We observed a different TE composition between the centromere region of Chr02, Chr03, Chr04, and Chr06 and that of other chromosomes, which may have led to their diverse evolutionary relationships.
- Ln 319, the large inversions -> large inversions
- Ln 321, near -> close to
- The end of Ln 322, lack a space before "To"
- Ln 329, were correlated with -> have been reported to regulate; also need citations here
- Ln 394, CIG42_01g0030700 -> and CIG42_01g0030700
- Ln 403, suggesting at -> suggesting
- Ln 415, the study -> our study
- Ln 557, near -> nearly; 6-bp -> a 6-bp; 18-bp -> an 18-bp; Ln 558, deletions -> deletion

Reviewer #4 (Remarks to the Author):

The manuscript of Zhang et al. reports on a pangenome of the genus Citrullus and shows how it is useful for genetic research in that species. I have seen the revised manuscript and the authors' response to the referees' comments on the initial submission. This is a data-rich manuscript. It is well-written, albeit dense in places. Many different topics are touched upon. All converge to show how useful the pangenome will be for applied genetics in water melon.

The authors have done a good job in addressing the referees' concern. The most critical comments came from referee 1. She/he questioned the novelty of the paper given the fact that another paper with a seemingly similar scope was published in Plant Biotechnology Journal recently. I do not share this concern. The Wu et al. paper has narrower taxon sampling and used an outdated sequencing method (PacBio CLR + Illumina sequencing). Important, the pangenomic analyses of Wu et al. were gene-based and their dataset had limited power to detect SV. The transcriptional and metabolomic analyses reported in the current paper add a dimension that is lacking in Wu et al. Zhang et al. go to much greater lengths to demonstrate the value of their resource in translational applications.

I recommend acceptance of this revised version.

Author Rebuttal, first revision:

Reviewer #2 (Remarks to the Author):

The authors have addressed all of the concerns that I raised in my initial review. The revisions have been executed with care and have substantially improved the manuscript.

The authors have made a significant effort to clarify the previous ambiguities, strengthen the methodology, and provide a more robust analysis of the data. This has greatly enhanced the overall quality and scientific rigor of the paper.

In terms of novelty and relevance, to the best of my knowledge the findings presented in the manuscript remain novel and contribute meaningfully to watermelon genomic resources.

I believe that the manuscript is now well-suited for publication in Nature Genetics.

A few minor comments:

line 329 the term correlated implies a quantitative analysis, if so mention the value.

Response:

Thank you for the comment. In response, we revised that sentence and added related citations.

line 330 how do these insights help understand the SVs? Their origin? maybe rephrase.

Response:

Thank you for pointing out the issue of understanding SVs. We rephrased it as providing valuable insights into understanding functions of these SVs.

line 451 again the use of strong correlation is vague - write a value.

Response:

Thank you for the suggestion. We added the value in the manuscript.

line 495 you mention correlation but the value in parenthesis is not the correlation coefficient but the r-square

Response:

Thank you for pointing out the error. We revised in the manuscript.

line 556 missing "be" in "seems to expressed" - seems to be expressed

Response:

Thank you for pointing out the error. We revised in the manuscript.

line 558 please write where upstream these deletions are located (e.g kb from translation start site)

Response:

Thank you for the suggestion. We added the position information of the two deletions in the manuscript.

Reviewer #3 (Remarks to the Author):

Many thanks for the opportunity to see the revision. The authors have appropriately addressed my concerns. While the manuscript has been improved, there are still some places that could benefit from the improvement of the use of English.

- Ln 4, is Bob Jarret really not Robert Jarret?

Response:

Thank you for your careful review of our manuscript. We appreciate your attention to detail. Regarding your query on Line 4, "Bob Jarret" should indeed be corrected to "Robert Jarret." We apologize for any confusion this may have caused and have made the necessary correction in the revised manuscript.

- Ln 26, perhaps changing to “by adding 399.2 Mb of sequences and 11,225 genes,”

Response:

Thank you for the suggestion. We have revised it in the manuscript.

- Ln 33, suggesting at -> suggesting

Response:

Thank you for the suggestion. We have revised it in the manuscript.

- Ln 138-139, change to: The development of PKR6 has introgressed several genes conferring resistance against xxxx

Response:

Thank you for the suggestion. We have revised it in the manuscript.

- Ln 220-224, if I understand correctly, change to: We observed a different TE composition between the centromere region of Chr02, Chr03, Chr04, and Chr06 and that of other chromosomes, which may have led to their diverse evolutionary relationships.

Response:

Thank you for your thoughtful suggestions. We agree with your suggestion for a more reasonable expression and have amended the manuscript according to your comment.

- Ln 319, the large inversions -> large inversions

Response:

Thank you for the suggestion. We have revised it in the manuscript.

- Ln 321, near -> close to

Response:

Thank you for the suggestion. We have revised it in the manuscript.

- The end of Ln 322, lack a space before “To”

Response:

Thank you for pointing out the error. We have revised it in the manuscript.

- Ln 329, were correlated with -> have been reported to regulate; also need citations here

Response:

Thank you for the comment. In response, we revised the sentence and added related citation in the manuscript.

- Ln 394, CIG42_01g0030700 -> and CIG42_01g0030700

Response:

Thank you for the suggestion. We have revised it in the manuscript.

- Ln 403, suggesting at -> suggesting

Response:

Thank you for the suggestion. We have revised it in the manuscript.

- Ln 415, the study -> our study

Response:

Thank you for the suggestion. We have revised it in the manuscript.

- Ln 557, near -> nearly; 6-bp -> a 6-bp; 18-bp -> an 18-bp; Ln 558, deletions -> deletion

Response:

Thank you for the suggestion. We have revised this sentence in the manuscript.

Reviewer #4 (Remarks to the Author):

The manuscript of Zhang et al. reports on a pangenome of the genus *Citrullus* and shows how it is useful for genetic research in that species. I have seen the revised manuscript and the authors' response to the referees' comments on the initial submission. This is a data-rich manuscript. It is well-written, albeit dense in places. Many different topics are touched upon. All converge to show how useful the pangenome will be for applied genetics in water melon.

The authors have done a good job in addressing the referees' concern. The most critical comments came from referee 1. She/he questioned the novelty of the paper given the fact that another paper with a seemingly similar scope was published in *Plant Biotechnology Journal* recently. I do not share this concern. The Wu et al. paper has narrower taxon sampling and used an outdated sequencing method (PacBio CLR + Illumina sequencing). Important, the pangenomic analyses of Wu et al. were gene-based and their dataset had limited power to detect SV. The transcriptional and metabolomic analyses reported in the current paper add a dimension that is lacking in Wu et al. Zhang et al. go to much greater lengths to demonstrate the value of their resource in translational applications.

I recommend acceptance of this revised version.

Response:

We sincerely appreciate your thorough evaluation of our manuscript and the positive feedback provided. Your recognition of the extensive efforts we put into addressing referee comments is invaluable.

Final Decision Letter:

4th Jun 2024

Dear Dr. He,

I am delighted to say that your manuscript "Telomere-to-telomere Citrullus Super-pangenome Provides Direction for Watermelon Breeding" has been accepted for publication in an upcoming issue of Nature Genetics.

Your paper will be published online after we receive your corrections and will appear in print in the next available issue. You can find out your date of online publication by contacting the Nature Press Office (press@nature.com) after sending your e-proof corrections.

Before your paper is published online, we shall be distributing a press release to news organizations worldwide, which may very well include details of your work. We are happy for your institution or

funding agency to prepare its own press release, but it must mention the embargo date and Nature Genetics. Our Press Office may contact you closer to the time of publication, but if you or your Press Office have any enquiries in the meantime, please contact press@nature.com.

Please note that *Nature Genetics* is a Transformative Journal (TJ). Authors may publish their research with us through the traditional subscription access route or make their paper immediately open access through payment of an article-processing charge (APC). Authors will not be required to make a final decision about access to their article until it has been accepted. Find out more about Transformative Journals

Authors may need to take specific actions to achieve compliance with funder and institutional open access mandates. If your research is supported by a funder that requires immediate open access (e.g. according to Plan S principles) then you should select the gold OA route, and we will direct you to the compliant route where possible. For authors selecting the subscription publication route, the journal's standard licensing terms will need to be accepted, including [a href="https://www.nature.com/nature-portfolio/editorial-policies/self-archiving-and-license-to-publish](https://www.nature.com/nature-portfolio/editorial-policies/self-archiving-and-license-to-publish). Those licensing terms will supersede any other terms that the author or any third party may assert apply to any version of the manuscript.

If you have not already done so, we invite you to upload the step-by-step protocols used in this manuscript to the Protocols Exchange, part of our on-line web resource, natureprotocols.com. If you

complete the upload by the time you receive your manuscript proofs, we can insert links in your article that lead directly to the protocol details. Your protocol will be made freely available upon publication of your paper. By participating in natureprotocols.com, you are enabling researchers to more readily reproduce or adapt the methodology you use. [Natureprotocols.com](https://natureprotocols.com) is fully searchable, providing your protocols and paper with increased utility and visibility. Please submit your protocol to <https://protocolexchange.researchsquare.com/>. After entering your nature.com username and password you will need to enter your manuscript number (NG-A63345R1). Further information can be found at <https://www.nature.com/nature-portfolio/editorial-policies/reporting-standards#protocols>

Sincerely,
Wei

Wei Li, PhD
Senior Editor
Nature Genetics
New York, NY 10004, USA
www.nature.com/ng